# Missing Value Uncertainty: Could Collecting Missing Values Change the Prediction?

## Abstract

In mission-critical domains such as sensor networks, operators often face the critical decision of whether to act on incomplete information or if collecting missing values is likely to change the prediction. Existing methods typically focus on imputing missing values or quantifying model uncertainty, but they do not directly assess the stability of a prediction if missing values were to be revealed. To address this gap, we introduce a framework for Missing Value Uncertainty (MVU), which is the distribution of predictions induced by incomplete inputs at inference time. We formalize the problem by defining *hard confidence*: the probability that a prediction will not change after collecting the missing data. We propose a novel Direct Missing Value (DMV) to efficiently estimate the MVU distribution, bypassing the need for expensive Monte Carlo sampling or retraining the model. Second, we introduce the Missing Value Calibration Error (MVCE), a new metric specifically designed to evaluate the calibration of hard confidence values, and a post-hoc calibration procedure to improve MVU estimation. We showcase our method and metric on synthetic and real-world datasets.

## 1 Introduction

In high-stakes domains such as healthcare and security, decisions are often made with incomplete information. This raises a critical and practical question for a human operator: **Is it worth the cost and effort to collect missing input values for a specific instance at inference time?** For example, a security analyst observing a potential threat with data from a partially failed sensor network must determine whether to act immediately or deploy resources to gather more information. Alternatively, a self driving car with a partially obscured camera will have to determine if the quality of image has dropped beyond where it can safely perform autonomous actions (ISO, 2024). We start to answer these questions by asking whether collecting missing values is likely to change the prediction. If additional information is unlikely to alter the optimal course of action, an operator can proceed without collecting missing values. However, if the missing values could significantly change the decision, the most prudent action is to collect them first. This paper centers on developing a framework that, given a partially observed input at inference time, can help an operator decide if collecting missing input feature values may change the decision.

Prior work has addressed aspects of this problem but fails to directly answer the central question of quantifying when more information may change the prediction. Missing value literature, for instance, primarily focuses on developing methods like imputation to handle missing values and make the best possible prediction given the observed values (Little & Rubin, 2019; Azur et al., 2011). While useful, these techniques do not quantify the uncertainty introduced by the missing values, leaving the operator unsure of how the prediction might change if the missing values were revealed.

Separately, research in uncertainty quantification has focused on a different question: "How likely is the prediction to be correct?". This question is useful for deciding whether to accept a model's prediction, corresponding to the notion of **soft confidence** that we explain in Section 3. However, it does not inform an operator about the stability of the prediction. Our work instead focuses on a notion called **hard confidence**: the probability that a prediction will *not* change if missing values are collected (see Section 3 for formal definition). Furthermore, existing work on epistemic uncertainty—i.e., uncertainty that can be reduced by

more information—typically addresses model uncertainty arising from limited training data (Liu et al., 2019). While this may inform a decision maker if more training data is needed, it is orthogonal to the uncertainty from missing values for a single test-time instance, where the solution is to collect more features, not more training data. Given all this, to our knowledge, *no prior work has formalized, estimated, or evaluated the uncertainty stemming specifically from missing values at inference time to aid in deciding whether collecting missing values may be beneficial.* This represents a critical gap, as it leaves a practical decision-making problem without a principled solution.

To fill this gap, we propose a complete framework for analyzing **Missing Value Uncertainty (MVU)**, the distribution of predictions induced by missing inputs. We formalize the decision-making problem through the lens of hard confidence, which directly quantifies prediction stability. We establish imputation and Monte-Carlo baselines for estimating MVU and propose a novel explicit method for estimating MVU: the **Direct Missing Value (DMV)** estimator, which is significantly more efficient by circumventing sampling of the missing values, particularly in high dimensions. To evaluate these methods, we develop the **Missing Value Calibration Error (MVCE)**, a new metric designed specifically for assessing the calibration of hard confidence values. Finally, we introduce a post-hoc procedure to improve the calibration of any MVU estimation method. Our main contributions are:

- We formalize the problem of missing value uncertainty (MVU) to aid operator decisions about collecting missing features at inference time.
- We develop DMV, a novel and efficient approach that directly estimates MVU without requiring expensive Monte Carlo sampling.
- We define a novel metric, the Missing Value Calibration Error (MVCE), for evaluating MVU estimation methods on any dataset and propose a MVCE-based post-hoc calibration approach that can improve an MVU method after training.
- We empirically validate our methods on both synthetic and real-world datasets.

## 2 Related Works

**Missing Values** Missing values are features of an input sample that are unobserved where having an observed value could be useful for prediction (Little & Rubin, 2019). This divides the sample $x$ into observed features $x_{\mathcal{O}}$ and missing features $x_{\mathcal{M}}$, which notably may be a different set of features per sample. Missing data is commonly divided into different types based on the mechanism of missingness. The simplest case of missing completely at random (MCAR) assumes no relationship between what features are missing and the values of any features (missing or observed). Missing at random (MAR) allows the missingness to be correlated to observed feature values, while missing not at random (MNAR) allows missingness to depend on the full range of values of both missing and observed features along with the labels (Little & Rubin, 2019; Bennett, 2001).

Most prior work focus on the same missingness setup between training and evaluation (Little & Rubin, 2019; Zhou et al., 2023). Within training data, missingness is common in settings where complete data is often not present, such as medical tests or survey data (Goldberger et al., 2000; Lewenberg et al., 2017; Ma et al., 2018). We however focus on the case where missingness exists in test data but not in training data; this is difficult as we cannot assume the mechanism in test data matches training data. A simple approach is to augment fully observed data to simulate missingness (Troyanskaya et al., 2001); we will call this "training masking", which is notably distinct from actual missing values as we can access ground truth for the missing values for evaluation. Some works further explore the concept of missingness shifts in a setting where both training data and testing data have missing data, but under different mechanisms of missingness (including two different setups under the same type) (Stokes et al., 2025; Zhou et al., 2023). Our paper focuses on missing values at *test time* due to device failures or collection problems, even if you have complete data at training time.

One approach to handling missing values is imputation (Little & Rubin, 2019; Azur et al., 2011; Khosravi et al., 2019), though this lacks a mechanism for estimating uncertainty and depending on the approach may require complete training data. Some imputation approaches can produce multiple samples conditioned on

the observation which can be leveraged through Monte Carlo to estimate uncertainty; image inpainting is a notable example for high dimensional spaces (Ma et al., 2018; Zhang et al., 2023; Liu et al., 2023). Other approaches modify the classifier to allow missing inputs often through masking training data, which may even be able to estimate missing value uncertainty directly (Khosravi et al., 2019), though these often impose restrictions on the model architecture or input space. When dependencies exist between the features that are missing and the values of features such as MNAR data, the locations of missing features can (and should) be leveraged as additional information about values of missing features (Steck, 2010; Bennett, 2001); however such improvements are not feasible in MCAR. While we leverage some of these methods, our proposed framework is not primarily concerned with predictions with missing values but with uncertainty estimation due to missing values.

**Active Learning** Active learning is a field in machine learning that seeks to answer a similar question to our objective: which feature or samples are most useful to label to improve the quality of our prediction (Settles, 2009). A problem in that field of particular relevance to our work is the active surveying problem; survey responses are modeled as a set of questions with incomplete answers, and the model must decide which question is most useful to ask next (Lewenberg et al., 2017; Ma et al., 2018). This setup naturally fits as a missing values problem, though notably work in this area is often focused on selecting features based on information in the feature distribution $p(X_{\mathcal{M}}|X_{\mathcal{O}})$ rather than measuring uncertainty in the prediction due to missing features $P(Y|X_{\mathcal{O}})$. Additionally, these approaches tend to terminate feature collection based on confidence in the prediction which may lead the model to continue to collect information beyond when additional information can improve the prediction.

**Uncertainty** Uncertainty estimation allows our model to not just make a prediction, but estimate its quality with respect to modeled sources of uncertainty. This makes it a natural fit for estimating the confidence in our handling of missing values. Uncertainty is generally broken into two types: aleatoric and epistemic. Aleatoric uncertainty is any source of variability outside our model (notably unmeasurable), which we simply model as random behavior. Epistemic uncertainty is a reducible form of uncertainty in the model due to a lack of data, which can be further decomposed based on the type of data; commonly parameter uncertainty is considered (Liu et al., 2019). For our work, we are particularly interested in uncertainty from missing values (MVU). While MVU is reduced as more missing values are revealed, i.e., $x_{\mathcal{O}} \to x$, it is not always possible to collect additional features, which may classify it as either aleatoric or epistemic depending on modeling assumptions. Some uncertainty work considers making the prediction robust to missing values (Zaffran et al., 2023), though they do not directly report MVU. Some prior works such as Bayesian Inference (Gelman et al., 1995), Evidential Deep Learning (Sensoy et al., 2018), and other second-order uncertainty approaches (Sale et al., 2023; Bengs et al., 2022) report prediction uncertainty, though this is limited to aleatoric or other types of epistemic such as parametric rather than our goal of uncertainty due to *missing values.*

**Conformal Prediction** Conformal prediction is an area that upon initial inspect appears to solve a similar problem to our goal, though closer inspection shows we are distinct. Conformal prediction is an uncertainty quantification method that can provide distribution-free, statistical guarantees on predictions for any underlying model (Vovk et al., 2005). Instead of a single prediction, it produces a prediction set (for classification) or interval (for regression) that is guaranteed to contain the true outcome with a user-specified probability (e.g., 90%), without depending on assumptions about underlying data distribution or model correctness (Angelopoulos & Bates, 2021). Recent work has adapted the area to missing values (Zaffran et al., 2023) and high-dimensional settings (Romano et al., 2019; Zaffran et al., 2023). However, these methods aim to keep other types of uncertainty in the prediction valid despite missing values, instead of analyzing the uncertainty added specifically *due to missing values* to determine if there is insufficient information.

**Model Calibration** While not directly related, calibration ideas inspired our MVU metric and thus we briefly explain them here. Calibration in machine learning refers to the alignment between a model's predicted probabilities and the true likelihood of outcomes, ensuring that confidence scores accurately reflect real-world chances of correctness. Expected Calibration Error (ECE) is a key metric used to assess the calibration of probabilistic classifiers by quantifying the difference between the predicted probabilities and actual outcomes (Naeini et al., 2015; Guo et al., 2017). Nixon et al. (2019) discuss the shortcomings of the ECE metric and introduces Adaptive Calibration Error (ACE) and Thresholded ACE (TACE) to address its limitations, particularly in multi-class settings. Vaicenavicius et al. (2019) build on this by proposing a general theoretical

framework for evaluating the calibration of probabilistic classifiers. Calibration properties of modern neural networks are analyzed by Minderer et al. (2021). Unlike previous calibration works, we aim to estimate the uncertainty due to missing values, rather than the overall uncertainty of the prediction. Our problem is distinct from standard calibration works.

**Relationship to Our Work**

Ultimately, prior methods do not fully address the fundamental problem of estimating and evaluating the uncertainty due to missing values at *test time*, in order to answer whether we need to collect additional information. Most prior methods focus on training data with missingness to predict in a *similar missingness setup* at test time, rather than working with clean training data for an *unknown missingness setup*. On the other hand, uncertainty and calibration works focus on aleatoric uncertainty or epistemic uncertainty due to limited training data—which results in uncertainty of estimated parameters. Our work lies at the intersection of missing values at test time that were not missing in training data and uncertainty due to these missing values. While there has been limited work on robustness with missing values at test time (Khosravi et al., 2019), these works impose very strong constraints (i.e., circuit constraints) on the component models to enable exact inference.

## 3 Soft and Hard Voting Classification Rules and Confidences

Given a distribution of predicted probabilities $p(\Phi)$, which implicitly represents a weighted ensemble of model predictions (Hansen & Salamon, 2002), we consider two decision rules: soft and hard voting (Kittler et al., 1998). **Soft voting** (sometimes called the sum rule or average rule) averages class probabilities directly and corresponds to the standard Bayes optimal classification rule. In contrast, **hard voting** (also called majority voting) averages the argmax predictions from each model, forming a robust ensemble classifier. While soft voting is well-studied, we focus on hard voting, which is underexplored in the area of missing values, as its resulting confidence value is more actionable for deciding whether to collect missing values. Though we apply this approach to missing value uncertainty, it may be of independent interest for other types of epistemic uncertainty. We first discuss these rules generically before specializing to the missing value case in the next section. Note that while typically the term "voting" references an ensemble setup, for our work we will apply it instead to a distribution of predictions, which can be either sampled or used directly to compute the desired queries.

**Notation** For discrete class distributions, we will often use $P(Y)$ (or conditional variants like $P(Y|X_{\mathcal{O}})$) to denote either the distribution itself or the vector of probabilities $[P(Y = 1), P(Y = 2), ..., P(Y = k)]$ for all $k$ classes, which fully defines the distribution. In many cases these probabilities are defined by a vector or parameters $\phi$ such that $P(Y = j) \equiv \phi_j$.

### 3.1 Background: Bayes Optimal Classification via Soft Voting

Given a distribution of predictions $p(\Phi)$ where $\Phi$ represent class probabilities, the soft-voting rule uses $\Phi$ directly as a soft vote and averages over the $p(\Phi)$ distribution:

$$y_{\text{soft}} \triangleq \arg\max_j \mathbb{E}_{p(\Phi)}[\Phi]_j \equiv \arg\max_j P(Y = j), \ c_{\text{soft}} \triangleq \max_j \mathbb{E}_{p(\Phi)}[\Phi]_j \equiv \max_j P(Y = j), \tag{1}$$

where $P(Y = j) = \mathbb{E}_{p(\Phi)}[P(Y = j; \Phi)] = \mathbb{E}_{p(\Phi)}[\Phi]_j$ is the marginal probability of $Y$ when marginalizing over the uncertainty in $p(\Phi)$. When properly calibrated, this soft voting classification is equivalent to the *Bayes optimal classification rule*. The Bayes rule is the most common way to classify because it minimizes the misclassification error. The confidence value $c_{\text{soft}}$ is simply the probability that the selected class is correct. When given to a human operator, they could use $c_{\text{soft}}$ to determine whether they should accept or ignore the prediction depending on the confidence level required. If the confidence is high, the operator could confidently accept the prediction. However, if the confidence is low (e.g., close to $1/k$), then the operator should simply ignore the prediction since it is uninformative. Thus, from a practical standpoint, it mostly provides useful and actionable information if the confidence is high. Additionally, since soft voting is only Bayes optimal when properly calibrated, relying on it for our confidence values may be vulnerable to approximation error suggesting the need for an alternative (Kuśmierczyk et al., 2020).

## 3.2 Cumulative Probability Classification via Hard Voting

We propose to use hard voting (i.e., majority voting) as an alternative and complementary classification rule that yields distinct information compared to soft voting confidence. In particular, we will explain why it is useful for deciding whether to collect more information or not. The hard-voting rule uses the argmax of $\Phi$ (i.e., a single class hard vote) and averages these hard votes over the $p(\Phi)$ distribution:

$$y_{\text{hard}} \triangleq \arg\max_j \mathbb{E}_{p(\Phi)}[\text{OneHotArgmax}(\Phi)]_j \equiv \arg\max_j P(Y_{\text{vote}} = j) \tag{2}$$

$$\equiv \arg\max_j \Pr(\bigcap_{j' \neq j} \Phi_j \geq \Phi_{j'}), \tag{3}$$

$$c_{\text{hard}} \triangleq \max_j \mathbb{E}_{p(\Phi)}[\text{OneHotArgmax}(\Phi)]_j \equiv \max_j P(Y_{\text{vote}} = j), \tag{4}$$

where $\text{OneHotArgmax}(\phi) \triangleq \text{OneHot}(\arg\max_j \phi_j)$ is the one-hot encoding of the argmax function and $Y_{\text{vote}} \triangleq \arg\max_j \Phi_j$ is a random variable corresponding to the hard vote of a single $\Phi$ (where each $\Phi$ intuitively represents a model in the ensemble). Note that the objective is equivalent to the cumulative probability of the distribution in a region defined by the inequalities $\bigcap_{j' \neq j} \Phi_j \geq \Phi_{j'}$. Another interpretation is that each model performs the Bayes optimal classification *locally* using its own belief and then we take an average over these local classifications. Because hard voting only considers the index of the largest probability, it ignores the weightings and thus is naturally more robust to miscalibrated model predicted probabilities or classifiers that do not output probabilities (Tumer & Ghosh, 1996; Suen & Lam, 2000).

**Theoretic Foundation** Hard voting has been studied in within the PAC-Bayesian framework, which attempts to bound the performance of simpler posterior distributions relative to stochastic model selection over the Gibbs distribution (McAllester, 2003). While previous works loosely bounded the risk of hard voting by twice the Gibbs risk, the C-bound shows a tighter bound on the risk which is shrunk by increased variance in the distribution (Germain et al., 2015). Thus, hard voting over the distribution $p(\Phi)$ is a theoretically optimal strategy that strictly benefits from the uncertainty in the distribution, even when the model is miscalibrated.

**Hard Voting Confidence Values For Deciding Whether to Collect More Information** In the context of epistemic uncertainty where the uncertainty could be reduced by collecting more information, the hard voting confidence values provide the probability that the prediction would stay the same even if all missing values were revealed. If the hard confidence is high, then gathering more information is unlikely to change the predicted class nor the soft confidence. If the hard confidence is low, then it means that more information could change the predicted class and improve soft confidence.

**Comparing Soft Voting and Hard Voting** Soft and hard voting confidences capture complementary information for decision-making: soft confidence helps determine whether to **accept a class prediction as likely correct**. When the soft confidence is low, hard confidence can determine how much of this uncertainty is tied to insufficient information. In other words, hard confidence informs the decision to **collect more information** to reduce epistemic uncertainty. A few distinctions are:

- *Behavior with No Uncertainty:* For a degenerate distribution $p(\Phi)$ where epistemic uncertainty is zero, the predictions are identical. However, soft confidence can vary between $1/k$ and 1, while hard confidence is always 1, correctly reflecting that no new information will change the outcome due to non-informative or dependent unobserved features.

- *Sensitivity to Variance:* Soft confidence depends only on the mean of the distribution $p(\Phi)$ and is insensitive to its variance. In contrast, hard confidence decreases as the variance increases, thereby capturing the spread of the epistemic uncertainty, not just its central tendency.

## 4 Missing Value Uncertainty (MVU)

Having discussed uncertainty from a generic distribution of predictions $p(\Phi)$, we now formalize the Missing Value Uncertainty (MVU) distribution, which arises from incomplete inputs at inference time. To distinguish MVU from standard epistemic uncertainty, we briefly contrast it with the uncertainty stemming from finite training data.

Standard epistemic uncertainty typically refers to model uncertainty from a finite training set $\mathcal{D}_n$. In a Bayesian context, this induces a posterior over model parameters, $p(\Theta|\mathcal{D}_n)$, which in turn creates a distribution of predictions. This uncertainty is reducible, as it vanishes in the limit of infinite training samples ($n \to \infty$). In contrast, MVU is an orthogonal form of epistemic uncertainty induced by missing values for a single test-time instance. It is reducible not by collecting more training data, but by observing the missing values of that specific instance. We now formalize this concept.

**Definition 1** (Missing Value Uncertainty Distribution)**.** *Given a joint distribution $p(X, Y)$, let us define the true class distribution $\pi(x)$ given complete inputs and the corresponding random variable $\Phi$ as:*

$$\pi(x) \triangleq P(Y|X = x), \qquad \Phi \triangleq \pi(X) \equiv \pi(X_{\mathcal{O}}, X_{\mathcal{M}}) \tag{5}$$

*where $\pi : \mathcal{X} \to \Delta^{|\mathcal{Y}|-1}$ is a deterministic function mapping a complete input to a probability vector on the simplex and $\Phi$ is the random variable representing these class probabilities given the random input $X$, which can be divided into random observed features $X_{\mathcal{O}}$ and missing $X_{\mathcal{M}}$. Given these, the Missing Value Uncertainty (MVU) distribution given an observed set of input values $X_{\mathcal{O}}$ (and set of missing input values $X_{\mathcal{M}}$) is defined as the conditional of $\Phi$ given our observation $p(\Phi|X_{\mathcal{O}} = x_{\mathcal{O}})$.*

While $\pi(X)$ is deterministic given $X$, the partial observation $x_{\mathcal{O}}$ means the MVU distribution still has randomness with respect to the unobserved $X_{\mathcal{M}}$ (there is no randomness when data is fully observed, giving $X_{\mathcal{M}} = \emptyset$. It should also be noted that the target MVU distribution $p(\Phi|X_{\mathcal{O}} = x_{\mathcal{O}})$ is well-defined for any joint distribution $p(X, Y)$. Furthermore, note that $\pi$ is the optimal probabilistic classifier given complete inputs since it directly gives the class distribution conditioned on a complete input $x$. Finally, compared to epistemic uncertainty induced by finite samples, this uncertainty is induced by incomplete or partial inputs. This uncertainty distribution becomes degenerate (i.e., perfectly certain) when all inputs are observed. Thus, it aligns naturally with the view of epistemic uncertainty that it becomes zero when all information is revealed.

**Soft and Hard Voting for MVU** Applying the voting rules from Section 3 to the MVU distribution provides distinct types of information. Soft voting corresponds to the standard Bayes optimal classification given the observed features: $\arg\max_j P(Y = j|X_{\mathcal{O}} = x_{\mathcal{O}})$. As previously discussed, the resulting soft confidence does not inform an operator whether collecting more inputs would be useful. Hard voting, however, directly addresses this problem. The hard confidence derived from $\arg\max_j P(Y_{\text{hard}} = j|X_{\mathcal{O}} = x_{\mathcal{O}})$ represents the probability that the prediction will *not* change if the missing values are revealed. A high hard confidence therefore suggests that the decision is stable and collecting more data is unnecessary, a critical insight for operators of sensor networks where acquiring more information is costly.

## 5 Methods for MVU Estimation

Having established hard voting confidence in Section 3 and defining missing value uncertainty in Section 4, we now consider ways to estimate the MVU distribution $p(\Phi|X_{\mathcal{O}})$ for a distribution of $X_{\mathcal{O}}$. To the author's best knowledge, there are no prior methods that focus on estimating MVU, thus we first propose two natural baselines based on missing value imputation and Monte Carlo estimates using generative models. After establishing these baselines, we then introduce our novel Direct Missing Value (DMV) approach for directly estimating the MVU distribution without requiring imputation or sampling.

### 5.1 Baseline MVU Methods

**Imputation with Simple Variance** Assuming our classifier is trained on complete data and only accepts complete inputs, we need a method to fill in the missing features to make a prediction. We can start with a simple imputation approach such as zero imputation or mean imputation (Little & Rubin, 2019). However, we still require a method to estimate missing value uncertainty. A simple heuristic is to call the model output the "prediction mean", and combine with a heuristic for variance, then use the method of moments to estimate a natural distribution's parameters. Simply choosing a constant variance other than 0 may not be valid for all predicted depending on the distribution class. A better heuristic would be to create a Dirichlet distribution by simply scaling the predicted probability $\phi$ by some constant $c$ to produce $\alpha = \phi \cdot c$; this approach guarantees any positive scaling constant will produce a valid distribution. To make this parameter more intuitive, we

can leverage the fact that Dirichlet variance cannot be greater than or equal to $\phi \cdot (1 - \phi)$, at which point the distribution becomes degenerate as $\alpha = 0$. Thus, we scale the max variance using a constant $s$ between 0 and 1 (exclusive). Using the method of moments, this leads to $\alpha = \phi \cdot (\frac{1}{s} - 1)$, where larger values of $s$ represent more variance and thus a less confident prediction. One flaw with the simple variance approaches is our uncertainty does not change with respect to the specific features that are missing; we get the exact same uncertainty for a fully observed input as an input that imputed those same values. This will lead to the model being overconfident under large numbers of missing values, and underconfident with fully observed data.

**Monte Carlo Approximation** A notable downside to the imputation baseline is we are required to use heuristics to estimate uncertainty; we wish to instead explore the space of multiple potential imputations of missing features. A simple MVU approach is to use Monte Carlo samples of the missing values given observed values, i.e., samples from $p(X_{\mathcal{M}}|x_{\mathcal{O}})$, mapped through $\pi(x_{\mathcal{O}}, x_{\mathcal{M}})$ to empirically estimate the MVU distribution. This is motivated by the fact the only randomness remaining in Def. 1 is due to $X_{\mathcal{M}}$, which we assume is correlated to $x_{\mathcal{O}}$. At a high enough number of samples, this will produce an approximation close to the true MVU distribution. The key challenge involves sampling from the missing values given an (arbitrary) set of observed values $x_{\mathcal{O}}$ and unknown missingness pattern. Lower-dimensional cases could model the generator using a multivariate normal distribution, which has a simple closed form conditional distribution; however, in very high-dimensional settings such as image data, this is a poor approximation. Thus, we propose leveraging prior work on image inpainting to model $p(X_{\mathcal{M}}|x_{\mathcal{O}})$ in higher dimension cases, specifically diffusion models (Zhang et al., 2023). Due to the cost of running the diffusion model, we can only produce a limited number of samples from $p(X_{\mathcal{M}}|x_{\mathcal{O}})$, which would likely have too much variance to make a reasonable estimate of hard confidence if we directly map the samples of $X_{\mathcal{M}}$ through $\pi(x_{\mathcal{O}}, x_{\mathcal{M}})$ to produce samples of $\Phi$ Instead, we use these samples of $X_{\mathcal{M}}$ to learn the parameters of a Dirichlet distribution, giving us a smoother distribution $\hat{p}(\Phi|x_{\mathcal{O}})$ that can quickly be sampled for estimating hard confidence.

However, even at a small number of samples, estimating the parameters is still very expensive as it requires sampling from high-dimensional conditional models, making it impractical for real-time predictions. This motivates our proposed method which is a much more efficient alternative that does not require sampling from high-dimensional distributions.

## 5.2 Direct Missing Value Uncertainty (DMV)

We propose a novel direct estimator of the MVU distributions based on minimizing the KL divergence (or equivalently the negative log likelihood (NLL)) between the true and estimated MVU distributions: $\arg\min_{\psi} \mathbb{E}_{X_{\mathcal{O}}}[\mathrm{KL}(p(\Phi|X_{\mathcal{O}}), \hat{p}_{\psi}(\Phi|X_{\mathcal{O}}))] \equiv \arg\min_{\psi} \mathbb{E}_{X_{\mathcal{O}}}[\mathbb{E}_{\Phi|X_{\mathcal{O}}}[-\log \hat{p}_{\psi}(\Phi|X_{\mathcal{O}}))]]$, where $\psi$ represents the model parameters. While at first this might seem like a standard problem, the challenge is that $\Phi$ is latent rather than observed. Thus, we cannot directly estimate the NLL given only samples from $X_{\mathcal{O}}$. However, if we know $\pi(x) \triangleq P(Y|X = x)$ from Equation (5), then we can convert this NLL objective to an objective that only requires complete training samples, i.e., samples with all features (proof in Appendix B.2).

**Proposition 1.** *Given the optimal probabilistic predictor $\pi$ from Equation (5) and any set of observed feature indices $\mathcal{O}$, the following holds, where $\gamma$ is a constant w.r.t. $\psi$ but does depend on $\pi$:*

$$\mathbb{E}_{X_{\mathcal{O}}}[\mathrm{KL}(p(\Phi|X_{\mathcal{O}}), \hat{p}_{\psi}(\Phi|X_{\mathcal{O}}))] = \mathbb{E}_{X_{\mathcal{O}}, X_{\mathcal{M}}}[-\log \hat{p}_{\psi}(\pi(X_{\mathcal{O}}, X_{\mathcal{M}}) \mid X_{\mathcal{O}})] + \gamma_{\pi} \,. \tag{6}$$

The right hand side of Proposition 1 gives a natural way to directly estimate the uncertainty distribution given only complete samples and an estimate of the optimal complete predictor $\pi$. Importantly, *this approach can directly estimate the uncertainty distribution while elegantly bypassing the need to do conditional sampling of missing values given observed values.*

**Estimating both the optimal classifier and the MVU distributions from training data.** Given the above result, we propose a natural two-stage approach to estimating MVU distributions. First, we estimate the optimal predictor $\pi$ using standard supervised learning on the complete dataset. Second, we plug-in this estimated predictor into the objective above to learn the final MVU predictor. At first glance, it may seem that you could simply minimize the linear combination of the standard cross entropy loss for $\hat{\pi}_{\theta}$ and the

objective above for $\hat{p}_\psi$:

$$\arg\min_{\theta,\psi} \left( \mathbb{E}_{X,Y}[\ell_{\mathrm{CE}}(\hat{\pi}_\theta(X), Y)] + \mathbb{E}_{X_\mathcal{O}, X_\mathcal{M}}[-\log \hat{p}_\psi(\hat{\pi}_\theta(X_\mathcal{O}, X_\mathcal{M}) \mid X_\mathcal{O})] + \gamma_{\hat{\pi}_\theta} \right), \tag{7}$$

where $\ell_{\mathrm{CE}}$ is the standard cross entropy loss, $\theta$ are the parameters for the predictor on complete inputs (i.e., no missing values) and $\psi$ are the parameters of the MVU predictor given observed inputs $X_\mathcal{O}$. However, Equation (7) would incorrectly ignore the fact that the constant in Proposition 1 depends on $\hat{\pi}_\theta$ and thus in this case would depend on $\theta$. Moreover, the $\gamma_\pi$ term is not possible to approximate since we do not know the density of $p(\Phi \mid X_\mathcal{O})$. Therefore, we propose a bi-level optimization problem that will be valid because the lower-level problem assumes that the corresponding upper level variables are fixed:

$$\min_{\theta,\psi} \mathbb{E}_{X,Y}[\ell_{\mathrm{CE}}(\hat{\pi}_\theta(X), Y)], \quad \text{s.t. } \psi \in \arg\min_{\tilde{\psi}} \mathbb{E}_{X_\mathcal{O}, X_\mathcal{M}}[-\log \hat{p}_{\tilde{\psi}}(\hat{\pi}_\theta(X_\mathcal{O}, X_\mathcal{M}) \mid X_\mathcal{O})]. \tag{8}$$

This bi-level problem avoids the previous issue and can be easily decomposed into a two stage optimization problem:[1] (1) Optimize a classifier $\hat{\pi}_\theta$ on the *complete data* via standard supervised learning, and then (2) optimize an uncertainty model $\hat{p}_\psi(\Phi \mid X_\mathcal{O})$ assuming that $\hat{\pi}_\theta$ is fixed leveraging Proposition 1. The beauty of this approach is that the optimization problems are completely decoupled. Notably, it is possible to use a large *pretrained* model for $\hat{\pi}_\theta$, including a foundation model.

**Theoretic Guarantee for DMV** The theoretic version of this bi-level optimization matches the true MVU distributions (proof in Appendix B.3), which establishes the theoretic guarantees for DMV.

**Proposition 2.** *Let the non-parametric version of Equation* (8) *be defined as:*

$$q^*, f^* \triangleq \arg\min_{q,f} \mathbb{E}_{X,Y}[\ell_{\mathrm{CE}}(f(X), Y)], \quad s.t.\ q \in \arg\min_{\tilde{q}} \mathbb{E}_{X_\mathcal{O}, X_\mathcal{M}}[-\log \tilde{q}(f(X_\mathcal{O}, X_\mathcal{M}) \mid X_\mathcal{O})],$$

*where $q$ and $f$ are general non-parametric functions. The optimal solution $q^*$ corresponds to the true MVU distributions, i.e., $q^*(\Phi|X_\mathcal{O}) = p(\Phi|X_\mathcal{O})$.*

**Approximation of $\hat{p}_\psi$** Our DMV approach could use any approximation for $\hat{p}_\psi$, provided it can both be sampled (to compute MVU and MVCE) and evaluate the log-likelihood of $\phi$ (for minimizing our loss function during training). In our experiments, we leverage the Dirichlet distribution as it has a known closed form density parameterized on strength parameters $\alpha$. We directly estimate $\alpha$ by a learnable function $g_\psi(X_\mathcal{O})$ given observed values $X_\mathcal{O}$, i.e., $\hat{p}_\psi(\Phi|X_\mathcal{O}) = p_{\mathrm{Dir}}(\Phi|\alpha = g_\psi(X_\mathcal{O}))$. $g_\psi$ can use any standard classification architecture, simply requiring switching the final activation function from producing probability values $\phi$ to strictly positive strengths $\alpha$ (e.g. if the final activation function was softmax, we can instead use exponential). We additionally need our architecture to handle both partially and fully observed inputs; for our experiments we augment the input with the mask information (making the corresponding change to the architecture input dimensions), then zero missing values. This allows the model to distinguish between missing features and observed zeroes. For example, if the inputs are 3 channel RGB images, then we can add a 4th mask channel, which contains 1s at anywhere that missing values were zeroed and 0s otherwise. Finally, we train this model using the DMV bi-level optimization problem described above. Note these specific approaches (Dirichlet approximation and mask channel) are not required to use the DMV bi-level optimization; it may for instance be useful to use a normal distribution approximation for a regression problem, or to skip the mask channel when considering corrupted inputs such that specific mutated features are unknown.

## 6 Evaluating MVU via Missing Value Calibration Error (MVCE)

As stated in the introduction, the key decision question is: "Is it worth it to collect missing values (for this specific test sample)?" The answer to this question depends heavily on the specific real-world problem context such as the cost of revealing missing values (e.g., doing a biopsy is significantly more expensive than collecting blood pressure) and the costs of being wrong (e.g., performing surgery if there is no cancer has high cost). To formally evaluate the decision problem, we would have to specify a decision process with all

---

[1]In Appendix D, we discuss a regularized bi-level problem that cannot be decoupled into two stages.

associated actions, costs/rewards, world environment, etc. Thus, instead of focusing on a particular scenario, we aim for a problem-agnostic evaluation of MVU models by asking an uncertainty calibration question w.r.t. *hard* confidence: "When my model predicts a *hard confidence* of $c\%$ on a partially observed input $x_{\mathcal{O}}$, is the prediction on the fully observed $x$ the same $c\%$ of the time?" In other words, hard confidence reflects how much the prediction matches the decision with full observations, while soft confidence reflects the match of the prediction to the ground truth. This is similar but distinct from the standard uncertainty calibration question corresponding to *soft* confidence which is: "When my model predicts a soft confidence of $c\%$, does it match the true class $c\%$ of the time?" Specifically, the hard confidence calibration gives the probability that the prediction will not change, while the soft confidence calibration gives the probability the class is correct. While soft confidence values can be naturally evaluated using the well-known Expected Calibration Error (ECE) (Naeini et al., 2015; Guo et al., 2017), evaluating hard confidence values for MVU requires some adaptation. See Appendix C.1 for more detail about the definition of ECE beyond its relationship to MVCE.

### 6.1 Missing Value Calibration Error for Hard Confidence Evaluation

Because hard confidence aims to quantify the probability that the prediction will stay the same, the key idea is that we will simulate the phenomena of revealing all missing values using complete training data. Intuitively, we simulate a partially observed input $x_{\mathcal{O}}$ by dropping features from the full input $x$ and then compare with the prediction on the complete $x$. Compared to ECE, we are not estimating whether the prediction is accurate but whether the prediction *changed* on average after revealing all missing values. As a reminder, the aim of hard confidences is to help operators know whether collecting missing values would be useful or not. Given this, like ECE, MVCE first partitions the dataset into bins $\mathcal{B}$ based on the *hard* confidence values $\hat{c}_{\text{hard},i}$.

**Definition 2** (Missing Value Calibration Error (MVCE)). *Given a dataset of labeled pairs, their computed hard predictions and hard confidences, and a partition of the dataset $\mathcal{B}$, MVCE is defined as:* MVCE $= \sum_{B \in \mathcal{B}} \frac{|B|}{|\mathcal{D}|} \left| \text{cons}(B) - \bar{c}_{\text{hard}}^{(\mathcal{O})}(B) \right|$, *where* $\bar{c}_{\text{hard}}^{(\mathcal{O})}(B) \triangleq \frac{1}{|B|} \sum_{i \in B} \hat{c}_{\text{hard},i}^{(\mathcal{O})}$ *is the average hard confidence on the (simulated) partial input* $x_{\mathcal{O},i}$ *in bin $B$ and the* consistency *of bin $B$ is defined as* $\text{cons}(B) \triangleq \frac{1}{|B|} \sum_{i \in B} \mathbb{1}(\hat{y}_i^{(\mathcal{O})} = \hat{y}_i)$, *where $\hat{y}_i^{(\mathcal{O})}$ is the prediction on the partial input $x_{\mathcal{O},i}$ and $\hat{y}_i$ is the prediction on the complete input $x_i$.*

The *consistency* term captures the idea of how often the prediction changed when all the missing values were revealed. This is the key difference for evaluating hard confidence values for MVU while the other parts are similar to ECE. Note that since consistency does not consider the true label, it will not evaluate whether the model is accurate; for best results MVCE should be employed alongside traditional accuracy and ECE to fully evaluate a model. Unlike confidence $\bar{c}_{\text{hard}}^{(\mathcal{O})}(B)$, consistency $\text{cons}(B)$ can be computed using either $y_{\text{soft}}$ and $y_{\text{hard}}$; for the fully observed prediction they are interchangeable. For our experiments, we use $y_{\text{soft}}$ as it is notably less impacted by variance than $y_{\text{hard}}$ leading to less noise in the results. Additionally, it makes more sense to quantify whether $y_{\text{soft}}$ may shift as that is prediction approach an operator is more likely to use.

**Relation to Other Metrics** Our MVCE metric isolates the evaluation of the hard confidence values for MVU. As such, it does not evaluate the accuracy or standard calibration of the classifier. For this, one can simply use standard metrics like accuracy and ECE to evaluate the performance and calibration of the classifier. This is similar to how a classifier may be accurate but not calibrated or calibrated but not accurate. In practice, we recommend using accuracy, ECE and our MVCE metric so that all aspects of the system can be properly evaluated, but we focus on the evaluation of hard confidence values in this paper.

### 6.2 Post-hoc Calibration of MVU via MVCE

Similar to prior post-hoc calibration methods, we propose a post-hoc calibration method to improve MVU distribution estimate after training using the MVCE metric. While traditional methods for first-order uncertainty calibration typically adjust the predicted class probabilities (or soft confidences in our context) (Bengs et al., 2022), our goal is to improve the estimate of hard confidences, which depend on the variance. Therefore, we must calibrate the predicted MVU distribution $p(\Phi | x_{\mathcal{O}})$ directly, which will *implicitly* calibrate our confidence values.

Our post-hoc MVU adjustment approach can be viewed as a type of post-processing of the estimated MVU distribution, i.e., $\hat{p}_{\psi,\lambda}(\Phi|x_{\mathcal{O}}) = \Omega(\hat{p}_{\psi}(\Phi|x_{\mathcal{O}}), \lambda)$, where $\Omega$ modifies the MVU distribution based on parameters $\lambda$, ideally by changing either the variance or entropy without changing the mean. As one example which we will use in experiments, when the MVU is a Dirichlet distribution, i.e., $\hat{p}_{\psi}(\Phi|x_{\mathcal{O}}) = p_{\text{Dir}}(\Phi|\alpha = g_{\psi}(x_{\mathcal{O}}))$, we can simply scale the predicted Dirichlet strengths by a positive scalar $\lambda$, i.e., $\hat{p}_{\psi,\lambda} = p_{\text{Dir}}(\Phi|\alpha = \lambda g_{\psi}(x_{\mathcal{O}}))$. This scaling constant helps in the case where the model on average predicts too much or too little uncertainty; other cases would require a more advanced approach which is left to future work. Thus, our post-hoc calibration approach can be defined as: $\lambda^*(\psi) = \arg\min_{\lambda} \text{MVCE}(\hat{p}_{\psi,\lambda}(\Phi|x_{\mathcal{O}}))$. While the objective is non-differentiable due to the binning of $\mathcal{B}$, we can use zero-th order optimization to choose $\lambda$ such as a simple grid search for low-dimensional $\lambda$ or Bayesian optimization approaches.

## 7 Experiments

**Objective** First, we aim to validate our missing value calibration error (MVCE) metric through synthetic data, showing that the metric correctly penalizes approaches to estimating MVU that differ from known ground truth uncertainty (which is unknown for real datasets). Next, we compare MVU baselines, DMV and post-hoc calibration on the real-world CelebA dataset for three binary classification problems (i.e., blonde hair, eyeglasses, and smiling), all using simulated masking for missingness. We show that DMV has comparable MVCE to expensive diffusion sampling baselines while being 100x or more faster. Since it is possible achieve low MVCE through always estimating low confidence with poor handling of missing values, we also compare the models prediction accuracy on data with missing values (abbreviated "Acc MV") to the prediction accuracy on clean data (abbreviated "Acc C"). Finally, we evaluate DMV compared to efficient baselines on multiclass classification using MNIST, CIFAR10, and StarcraftCIFAR10, demonstrating it outperforms other approaches in estimating MVU.

**Common Implementation** We implement DMV (subsection 5.2) using a ResNet18 classifier, with the input layer augmented to include a mask channel and the final activation function swapped to output Dirichlet weights, representing $\hat{p}_{\psi}(\Phi|X_{\mathcal{O}})$ as $p_{\text{Dir}}(\Phi|\alpha = g_{\psi}(X_{\mathcal{O}}))$. A similar ResNet18 classifier is used for baseline classifiers, though they are trained using standard cross-entropy loss on clean data with the standard soft max activation function, making it a model for $\pi(X)$ instead. For most methods, we directly estimated the parameters of a Dirichlet distribution, which we then used with hard voting (subsection 3.2) via 1000 sample Monte Carlo to compute hard confidences for MVCE. Due to the randomness in sampling and mask generation, we ran multiple trials of each experiment and average the results. Appendix E contains additional details on experiments along with additional results including standard deviations. Finally, we implement calibration by directly scaling the Dirichlet parameters using a constant selected through a grid search of calibration constants selected through minimizing MVCE on validation data. Reported MVCE scores include the uncalibrated model MVCE and the calibrated model MVCE on test data.

**Validating MVCE Metric With Synthetic Data** To briefly validate our MVCE metric, we create a synthetic dataset where $X$ is sampled from a bivariate normal distribution, and randomly set one of the inputs to be missing at test time. As a multivariate normal, it is trivial to condition the distribiton to constructor our ground truth generator $p^*(X_{\mathcal{M}}|x_{\mathcal{O}})$. We then compare the ground truth generator to various modified generators, and are able to show that MVCE is increased when the generator differs from the ground truth. Additionally, we show that calibration can make the modified generators MVCE closer to ground truth, while ground truth generator is already calibrated. This matches our intuition that properly modeling the missing values leads to the best estimate of when the prediction may change given more information. For full details and results of this experiment, see Appendix E.4.

**Comparing Diffusion-Based, Scaled Max Variance and DMV Methods on CelebA Dataset** We now compare MVU methods on the real-world CelebA-HQ dataset (Karras et al., 2017). Since CelebA-HQ lacks attributes for classification, we obtain them through CelebAMask-HQ (Lee et al., 2019). The task is to predict three semantic attributes which are mostly on the top or bottom: Blond Hair (visible primarily in the top), Eyeglasses (partially visible in both halves), and smiling (primarily in the bottom). For missing data, we mask out either the top half or bottom half of the image; reported MVCE is the average MVCE between evaluating the test dataset on each mask separately. An real example of this sort of masking in the real world

would be during 2020 where most people wore face masks in public places, covering a significant amount of of the face that was available in training. This simple setup allows us to intuitively understand when the MVU should be high vs low, e.g., if the bottom is masked, then smiling should be relatively uncertain compared to the top being masked.

*Method Details:* For the Monte Carlo baseline, we make use of a pretrained CoPaint model (Zhang et al., 2023) that can inpaint CelebA-HQ images. We use CoPaint to approximate the generator $p(X_{\mathcal{M}}|x_{\mathcal{O}})$ in order to predict different possibilities for Monte Carlo sampling, (see subsection 5.1). The samples are passed through independently trained ResNet18 classifiers (He et al., 2016) trained to predict the three semantic attributes. We then use the prediction samples to estimate the parameters of a Dirichlet distribution. We additionally trained a separate DMV model for each feature which directly estimate the parameters, as described in "Common Implementation" above. We train DMV using a randomly selected masks from top missing, bottom missing, both missing, or neither missing. We also use a simple mean imputation baselines with three methods for estimating variance from the mean: 0, 0.5 max variance, and directly scaling the probability values by 10 (subsection 5.1). For post-hoc calibration (subsection 6.2), we evaluated the whole dataset on each mask, then selected the minimum average to select a single calibration constant for each method between 9 different choices.

*Results:* As seen in Table 1, exploring more possibilities by taking more diffusion samples leads to both improved mutated accuracy and confidence estimates. However, taking a large number of samples ends up being too slow to practically deploy. DMV produced notably faster results than the diffusion approach even at low sample counts, with MVCE comparable to that of the diffusion approach. While in some cases the mean imputation heuristic could produce similar performance, this usually comes at a tradeoff between minimizing MVCE and maximizing accuracy under missing values; DMV does not require such a tradeoff.

Table 1: On CelebA data, high number of diffusion samples produce the best results, but this takes too long to use in practice. DMV provides comparable MVCE while running much more efficiently than even single sample baselines, and has high accuracy across all features unlike the mean imputation approach. While calibration did not impact the ratings, the decrease suggests hard confidence is more useful for estimating whether collecting features may change the prediction. Note in a couple cases with the two best methods (diffusion 30 samples and DMV) calibration did lead to worse MVCE but the increase was numerically less than most methods improved; the small drop in accuracy and the already low MVCE suggest an already low variance leading to constants producing differences in MVCE within the margin of randomness. Reported times are for a single sample on the blond hair experiment (other attributes ran in comparable times).

| CelebA Method \ Missing | Time (sec.) | Feature: **Blond Hair** | | | | Feature: **Smiling** | | | | Feature: **Eyeglasses** | | | |
|---|---|---|---|---|---|---|---|---|---|---|---|---|---|
| | | Acc C | Acc MV | MVCE | MV Calib. | Acc C | Acc MV | MVCE | MV Calib. | Acc C | Acc MV | MVCE | MV Calib. |
| Diffusion - 1 Sample, 0 Variance | 182 | 93.8% | 92.1% | 0.0522 | - | 91.9% | 84.1% | 0.1357 | - | 99.3% | 99.0% | 0.0058 | - |
| Diffusion - 1 Sample, 0.5 Max Variance | 182 | 93.8% | 92.1% | 0.0328 | 0.0296 -10% | 91.9% | 84.1% | 0.1028 | 0.0990 -4% | 99.3% | 99.0% | 0.0045 | 0.0040 -11% |
| Diffusion - 1 Sample, Scale Probability | 182 | 93.8% | 92.1% | 0.0448 | 0.0326 -27% | 91.9% | 84.1% | 0.1226 | 0.1028 -16% | 99.3% | 99.0% | 0.0058 | 0.0045 -22% |
| Diffusion - 3 Samples - Empirical Variance | 547 | 93.8% | 92.4% | 0.0205 | 0.0176 -14% | 91.9% | 85.6% | 0.0555 | 0.0530 -5% | 99.3% | **99.0%** | 0.0037 | 0.0037 -1% |
| Diffusion - 30 Samples - Empirical Variance | 5465 | 93.8% | **93.2%** | **0.0062** | **0.0063** 2% | 91.9% | **86.4%** | **0.0163** | **0.0144** -12% | 99.3% | **99.0%** | 0.0021 | 0.0033 54% |
| Mean Imputation, 0 Variance | 0.547 | 93.8% | 89.4% | 0.0775 | - | 91.9% | 70.5% | 0.2633 | - | 99.3% | 97.5% | 0.0220 | - |
| Mean Imputation, 0.5 Max Variance | 0.538 | 93.8% | 89.4% | 0.0581 | 0.0548 -6% | 91.9% | 70.5% | 0.2335 | 0.2281 -2% | 99.3% | 97.5% | 0.0111 | 0.0109 -2% |
| Mean Imputation, Scale Probability | 0.619 | 93.8% | 89.4% | 0.0704 | 0.0583 -17% | 91.9% | 70.5% | 0.2548 | 0.2335 -8% | 99.3% | 97.5% | 0.0192 | 0.0113 -41% |
| Direct Missing Value (DMV) | 0.901 | 94.5% | **93.0%** | 0.0286 | 0.0059 -79% | 92.8% | **77.1%** | 0.0711 | 0.0695 -2% | 99.5% | **99.1%** | 0.0029 | 0.0036 23% |

**Comparison of DMV to Computationally Efficient Missing Value Approaches on Multiclass Datasets** While DMV managed to produce comparable performance to the diffusion model on CelebA data, the binary classification task lead to some misleading high performance of the naive baselines such as mean-imputation. Thus, to fully show the benefit of DMV, we test on three different 10-class datastes. MNIST (Deng, 2012) contains hand-drawn digits, with class labels for the 10 digits. CIFAR10 (Krizhevsky et al., 2009) contains low resolution images belonging to various classes, such as cat, airplane, or truck. StarCraftCIFAR10 (Kulinski et al., 2023) contains spacial data representing the location of units and environment features, and asks us to predict the map out of 5 options and whether the game is in the start or end. For all three datasets, we masked them by dividing the image into a 4x4 grid of "sensors", and had an independent random chance for each sensor to be dropped during evaluation, making the experiment MCAR during evaluation.

*Method Details:* Since the Monte Carlo baseline was too expensive to practically deploy, we skipped it for the multi-class datasets. We instead compare DMV (see "Common Implementation") to two baselines for handling missing values: clean classifier with mean imputation (using the training set mean), and a classifier robust to missing values (Resnet18 with the input augmentation of DMV but not the output, and trained using standard cross-entropy loss over masked data). Both DMV and the robust to missing classifier use the evaluation mask setup for training, making it MCAR. Both baselines used three methods for estimating variance: 0, 0.5 max variance, and directly scaling the probability values by 10 (subsection 5.1). For post-hoc calibration (subsection 6.2), we used the same mask setup on validation data to select between 9 different calibration constants based on minimizing MVCE.

*Results:* As shown in Table 2, DMV was able to consistently perform well across all three datasets, while the performance of the other approaches was far less consistent. Mean imputation notably only produced acceptable accuracy on StarCraft, regardless of variance approach. Meanwhile, a classifier simply robust to missing values depends on the variance heuristic matching the accuracy drop. It was notably simple to match these on MNIST as the low drop in accuracy allowed a low estimate of uncertainty to reduce MVCE. This suggests that in settings where the missing features are uninformative, it is sufficient to simply create a robust classifier. However, especially if the number of missing features may vary, estimating MVU using DMV allows identifying when the prediction is no longer usable due to too many missing features.

Table 2: The Direct Missing Value approach performance well across all three datasets, having low MVCE without a significant drop in accuracy. On MNIST, the drop in accuracy from clean to missing values was comparatively small on both approaches trained with masking leading to a simple heuristic for uncertainty giving minimal MVCE. This suggests that in cases where prediction is viable with limited features, it is better to devote training time to robustness over missing value uncertainty.

| Method \ Missing | MNIST | | | | | CIFAR10 | | | | | StarCraftCIFAR10 | | | | |
|---|---|---|---|---|---|---|---|---|---|---|---|---|---|---|---|
| | Acc C | Acc MV | MVCE | MV Calib. | | Acc C | Acc MV | MVCE | MV Calib. | | Acc C | Acc MV | MVCE | MV Calib. | |
| Mean Imputation, 0 Variance | 99.5% | 58.7% | 0.4116 | - | - | 71.8% | 32.1% | 0.6471 | - | - | 79.9% | 70.5% | 0.2231 | - | - |
| Mean Imputation, 0.5 Max Variance | 99.5% | 58.6% | 0.1900 | 0.1644 | -14% | 71.8% | 32.4% | 0.3092 | 0.2855 | -8% | 79.9% | 70.6% | 0.0493 | 0.0214 | -57% |
| Mean Imputation, Scale Probability | 99.5% | 59.0% | 0.3053 | 0.1903 | -38% | 71.8% | 32.0% | 0.4776 | 0.3115 | -35% | 79.9% | 70.6% | 0.0989 | 0.0233 | -76% |
| Missing Robust, 0 Variance | 99.6% | 96.9% | 0.0296 | - | - | 82.3% | **74.7%** | 0.1604 | - | - | 71.8% | 71.3% | 0.1278 | - | - |
| Missing Robust, 0.5 Max Variance | 99.6% | **96.9%** | **0.0031** | **0.0031** | 0% | 82.3% | 74.7% | 0.0804 | 0.0740 | -8% | 71.8% | 71.2% | 0.0809 | 0.0582 | -28% |
| Missing Robust, Scale Probability | 99.6% | 96.9% | 0.0168 | 0.0048 | -71% | 82.3% | 74.7% | 0.1349 | 0.0807 | -40% | 71.8% | 71.2% | 0.0937 | 0.0571 | -39% |
| Direct Missing Value | 98.5% | 92.7% | 0.0736 | 0.0354 | -52% | 80.0% | 72.3% | **0.0455** | **0.0455** | 0% | 79.8% | **78.5%** | **0.0201** | **0.0113** | -44% |

**Ablation Study on Alternative Mutators for DMV Training** For the majority of our experiments, we both trained DMV and evaluated MVU using a MCAR mutator. This choice maximizes entropy from missing values during training to minimize the chance of the model treating the absence of missing features itself as informative. We wish to determine whether DMV can be trained to leverage MNAR information, and wish to validate our hypothesis that MCAR produces a more robust model in absence of knowledge about the test time mutator. For this experiment, we focus on the Starcraft dataset, and construct two MNAR mutators: MNAR (high) which is more likely to drop regions with units, and MNAR (low) which is more likely to drop regions with no units. We reuse the MCAR mutator from previous experiments for comparison.

*Method Details:* We train three DMV models (see "Common Implementation"), each on a different one of the three mutators: MCAR, MNAR (high), and MNAR (low). At test time, each of these DMV models is evaluated on each of the three mutators to compare accuracy and MVCE.

*Results:* As shown in Table 3, all three models had comparable accuracy on clean data. When testing under each of the three mutators, the model with the best accuracy was the model trained on the same type of mutator. For both MNAR mutators, we also see the model with the worst performance was the model trained on the opposite MNAR mutator, while the model trained on MCAR produced reasonable accuracy and MVCE across all mutators. This suggests that if you have information on the mechanism of missingness that will be present at test time, you can leverage it to improve the model. However, if the mechanism of missingness is unknown, using a MCAR mutator will produce the best average case.

Table 3: Comparing DMV models trained on MCAR and on two MNAR setups on the Starcraft dataset, we see that the best average case performance is on the model trained on MCAR. Both MNAR setups performed better when evaluated under the matching mutator, and worse when evaluated under the opposite mutator. MNAR (high) had a higher chance of dropping regions with units, while MNAR (low) had a higher chance of dropping regions without units. For each mutator, the best accuracy and MVCE are bolded, and the second best underlined.

| Training Mutator | Clean | Test Time Missingness | | | | | |
| | | MCAR | | MNAR (high) | | MNAR (low) | |
| | Acc | Acc | MVCE | Acc | MVCE | Acc | MVCE |
|---|---|---|---|---|---|---|---|
| MCAR | 79.78% | **79.78%** | **0.0201** | 75.09% | 0.0863 | 78.86% | **0.0090** |
| MNAR (high) | 80.18% | 76.51% | 0.0412 | **76.18%** | **0.0529** | 76.49% | 0.0259 |
| MNAR (low) | 80.63% | 79.25% | 0.0407 | 71.43% | 0.1358 | **80.17%** | 0.0102 |

## 8 Discussion

**Value of Considering MVU** In the multiclass datasets for MNIST, training a model using mutated data is sufficient to prevent a significant drop in accuracy under missing values, allowing a simple heuristic for MVU and producing minimal MVCE. If the observed features at test time are expected to still have sufficient information (i.e. highly dependent features) and there are limits on how many can be drooped such that our confidence would never be too low, then it would be sufficient to simply prioritize robustness to missing values and not worry about estimating MVU. The value of MVU comes when we expect features providing essential information may be dropped, from either high drop rates or mostly independant features. As not all features are equally informative, MVU allows us to estimate if the specific features dropped contained information likely to change the prediction. It would be worth further exploring settings with low numbers of features (such as a higher drop rate on MNIST) and settings with more distinct mutators between training and testing time (to ensure confidence estimates are robust to a diverse range of possible mutators).

**Cost of Considering MVU** Since the heuristics for estimating MVU only produced low MVCE when the model was already robust, properly considering MVU required training a model to directly estimate the distribution. In DMV, we prioritized estimates of MVU over accuracy, though a modified loss function or an additional stage of training could be used to minimize the accuracy drop. Alternatively, splitting the model into a robust classifier and a confidence estimator could enforce no accuracy loss; for a Dirichlet this would be modeled as $\alpha = \phi \cdot c$, where $\phi$ is the output of your robust classifier and $c$ is from a confidence model. Thus, in a sufficiently trained model there only tradeoff for considering MVU is if there is a limited training budget, where it becomes necessary to weigh the value of training for robustness against the value of training to estimate confidence. Analyzing this tradeoff when cost of training is a concern is left up to future work.

**Missing Training Data** For our work, we only considered missingness in evaluation data and masking during training; we assume the training data to be fully observed. This is consistent with a common assumption in Active Feature Acquisition (Aronsson et al., 2025). This notably limits the applicability of our work to setups with built-in missingness; for example, a doctor with a patient's initial lab results may need to decide whether this information is sufficient to make a diagnosis or if more costly and invasive tests are required. While potentially limiting, since we assume the ability to collect features at test time we believe the setup where fully observed data can be obtained at training time is realistic such as the sensor network example (if malfunctions are expected at test time), autonomous vehicles (with partially obscured sensors due to weather), or facial images (that are physically masked). Future work will look to incorporate missing features in the dataset, notably adapting Missing Value Calibration Error and Direct Missing Value which both currently rely on fully observed features being available.

**Cost Sensitive Decision Rule** Within our work, we considered the value of collecting missing features as how likely the prediction is to change. However, its possible that some predictions may have a higher cost of misclassifications, which can be leveraged to produce a better estimate of uncertainty. This would impact the formulation of confidence in our work, which propagates to how we compute and calibrate MVCE. We further formulate this future direction in Appendix A.

# 9 Conclusion

In this paper, we addressed the challenge of whether it is worth the cost of collecting missing inputs based on information at inference time. We developed the concept of missing value uncertainty (MVU) to estimate hard confidence over the distribution of missing values. This can be used by an operator to estimate whether collecting missing values will change the prediction. We proposed a Direct Missing Value method to estimate MVU, and developed the MVCE metric to evaluate whether high MVU correctly corresponds to a change in prediction. Finally, we tested our methods on several real-world datasets, comparing different approaches for estimating MVU to show the benefit of DMV over the baselines. Our work overall is not meant to replace existing approaches to estimating uncertainty, rather we recommend using it alongside existing epistemic and aleatoric uncertainty methods.

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

The supplementary material is organized as follows:

- **Appendix A - Cost-Sensitive Hard Voting Classification Rule:** Extension of hard confidence that encodes distinct costs of wrong decisions rather than all wrong options being equal.

- **Appendix B - Proofs:** Additional proofs that were not essential in the main paper.

- **Appendix D - Extension of Bi-Level DMV Problem with Regularization:** An extension of the bi-level optimization approach from Section 4.2.

- **Appendix E - Additional Experimental Results:** Additional experimental results.
    - **Appendix E.1 - CelebA:** Extended results for the CelebA dataset, including reports of consistency, MVCE after calibration, standard deviations for experiments, and information on the diffusion model.
    - **Appendix E.2 - Multiclass Datasets:** Additional information for the MNIST, CIFAR10 and StarCraftCIFAR10 datasets, notably including standard deviations for experiments.
    - **Appendix E.4 - Synthetic:** Full details on the synthetic dataset and how it was generated.

## A  Cost-Sensitive Hard Voting Classification Rule

Within our work, we considered the value of collecting missing features as how likely the prediction is to change. However, its possible that some predictions may have a higher cost of misclassifications, which can be leveraged to produce a better estimate of uncertainty. The ensembling classification framework above can generalized to the case where different misclassifications have different costs defined by a cost function: $\ell(y, \hat{y})$ where $y$ is the true label and $\hat{y}$ is the predicted label. For example, in medical tests, false positives may have a much lower cost than false negatives. Thus, we generalize the local Bayes optimal classification rule used to generate each hard vote using a minimum cost classification rule: $y = \arg\min_j \mathbb{E}_{p(Y;\phi)}[\ell(Y, j)]$, where $p(Y; \phi)$ denotes the categorical distribution with parameter $\phi$. The Bayes optimal classification rule used in the previous sections can be seen as minimizing the 0-1 loss function: $y = \arg\min_j \mathbb{E}_{p(Y;\phi)}[\ell_{0-1}(Y, j)] = \arg\max_j \phi_j$. For a simple generalization, we can use the cost-sensitive loss that has different costs for misclassification based on the true class: $\ell_w(y, \hat{y}) \triangleq w_y \mathbb{1}(y \neq y')$, where $w_y$ is the cost associated with misclassifying a sample from class $y$. This yields the following cost-sensitive classification rule: $y = \arg\min_j \mathbb{E}_{p(Y;\phi)}[\ell_w(Y, j)] = \arg\max_j w_j \phi_j$. Intuitively, this changes the classification boundary on the simplex from being in the center (corresponding to a simple argmax) to being off the center depending on $w$. When used in the hard voting method to determine the votes, this cost-sensitive classification rule will produce a different predictions and confidences:

$$y_{\text{hard}}^{(w)} \triangleq \arg\max_j \mathbb{E}_{p(\Phi)}[\text{OneHotArgmax}(w \odot \Phi)]_j \equiv \arg\max_j p(Y_{\text{vote}}^{(w)} = j), \tag{9}$$

$$\equiv \arg\max_j \Pr(\bigcap_{j' \neq j} w_j \Phi_j \geq w_{j'} \Phi_{j'}), \tag{10}$$

$$c_{\text{hard}}^{(w)} \triangleq \max_j \mathbb{E}_{p(\Phi)}[\text{OneHotArgmax}(w \odot \Phi)]_j \equiv \max_j p(Y_{\text{vote}}^{(w)} = j). \tag{11}$$

The definition of MVCE (Section 6.1) can then naturally be generalized to the case of cost-sensitive classification by using the corresponding cost-sensitive predictions and confidence values. In this extension, we simply use the cost-sensitive hard confidence values to select the bins and to estimate the average hard confidence. When working with the MVCE metric for calibration (Section 6.2), we may wish to calibrate with respect to multiple cost functions to minimize bias in the calibration, which changes the objective to an expectation of MVCE with respect to a distribution of cost functions.

This cost-sensitive hard voting can also be further generalized by changing the cost function to $\ell(a, \hat{y})$ where $a \in \mathcal{A}$ represents an action we can take rather than simply a predicted label. Intuitively, instead of predicting a class, we are predicting a response to the class where some responses may have lower cost when uncertain. As an example, in the sensor network problem instead of simply identifying whether something is a threat, actions can represent different methods of dealing with the threat, ignoring it, or gathering information as the cost of each of these varies with whether the detected object is actually a threat.

# B  Proofs

## B.1  Miscellaneous Proof(s)

Proof of the equivalence between 0-1 loss minimization and Bayes optimal classification rule:

$$y = \arg\min_j \mathbb{E}_{p(Y;\phi)}[\ell_{0-1}(Y,j)] \tag{12}$$

$$= \arg\min_j \mathbb{E}_{p(Y;\phi)}[\mathbb{1}(Y \neq j)] \tag{13}$$

$$= \arg\min_j \sum_{j' \neq j} \phi_{j'} \tag{14}$$

$$= \arg\min_j 1 - \phi_j \tag{15}$$

$$= \arg\max_j \phi_j . \tag{16}$$

Proof that cost-sensitive classification is a weighted version of the Bayes optimal classification rule:

$$y = \arg\min_j \mathbb{E}_{p(Y;\phi)}[\ell_w(Y,j)] \tag{17}$$

$$= \arg\min_j \mathbb{E}_{p(Y;\phi)}[w_Y \mathbb{1}(Y \neq j)] \tag{18}$$

$$= \arg\min_j \sum_{j' \neq j} w_{j'} \phi_{j'} \tag{19}$$

$$= \arg\min_j \sum_{j'} w_{j'} \phi_{j'} - w_j \phi_j \tag{20}$$

$$= \arg\min_j -w_j \phi_j \tag{21}$$

$$= \arg\max_j w_j \phi_j . \tag{22}$$

## B.2  Minimizing DMV Objective

We prove that minimizing the DMV objective from subsection 5.2 is equivalent to an objective that only requires complete samples, i.e., samples in the training data have all the features

*Proof.* We can simply use the definition of KL divergence along with LOTUS to derive the result:

$$\mathbb{E}_{p(X_{\mathcal{O}})}[\mathrm{KL}(p_f(\Phi \mid X_{\mathcal{O}}), \hat{p}_\psi(\Phi \mid X_{\mathcal{O}}))] \tag{23}$$

$$= \mathbb{E}_{p(X_{\mathcal{O}})}[\mathbb{E}_{p_f(\Phi|X_{\mathcal{O}})}[-\log \hat{p}_\psi(\Phi \mid X_{\mathcal{O}}))]] + \gamma_f \tag{24}$$

$$= \mathbb{E}_{p(X_{\mathcal{O}})}[\mathbb{E}_{p(X_{\mathcal{M}}|X_{\mathcal{O}})}[-\log \hat{p}_\psi(f(X_{\mathcal{O}}, X_{\mathcal{M}}) \mid X_{\mathcal{O}})]] + \gamma_f \tag{25}$$

$$= \mathbb{E}_{p(X_{\mathcal{O}}, X_{\mathcal{M}})}[-\log \hat{p}_\psi(f(X_{\mathcal{O}}, X_{\mathcal{M}})|X_{\mathcal{O}})] + \gamma_f , \tag{26}$$

where (24) is by the definition of KL where $\gamma_f = \mathbb{E}_{p(X_{\mathcal{O}})}[\mathbb{E}_{p_f(\Phi|X_{\mathcal{O}})}[\log p_f(\Phi|X_{\mathcal{O}})]]$, (25) is by the law of the unconscious statistician (LOTUS), and the last is simply by combining the distributions. Importantly, note that $\gamma_f$ does depend on $f$, and thus $f$ must be fixed in the optimization problem for $\gamma_f$ to be a constant. □

## B.3  Optimal solution to non-parameteric problem

We prove the theoretic version of our bi-level optimization from Section 4.2 would result in the true missing value uncertainty

*Proof.* Since the upper problem is decoupled, this is simply standard cross entropy minimization, which is equivalent to KL divergence minimization between $f(X)$ and $p(Y|X)$. Thus, the non-parametric $f(X)$ if solved perfectly will be equal to $p(Y|X)$, i.e., $p_{f^*}(Y|X) = p(Y|X)$.

Given that $f^*(X) = p(Y|X)$, we then invoke Proposition 1 on the lower level problem:

$$\arg\min_{\tilde{q}} \quad \mathbb{E}_{X_\mathcal{O}, X_\mathcal{M}} \left[ -\log \tilde{q}(f^*(X_\mathcal{O}, X_\mathcal{M}) \mid X_\mathcal{O}) \right] \tag{27}$$

$$= \arg\min_{\tilde{q}} \quad \mathbb{E}_{X_\mathcal{O}} \left[ \mathrm{KL}(p_{f^*}(\Phi|X_\mathcal{O}), \tilde{q}(\Phi|X_\mathcal{O})) \right] \tag{28}$$

$$= \arg\min_{\tilde{q}} \quad \mathbb{E}_{X_\mathcal{O}} \left[ \mathrm{KL}(p(\Phi|X_\mathcal{O}), \tilde{q}(\Phi|X_\mathcal{O})) \right] \tag{29}$$

$$= p(\Phi|X_\mathcal{O}), \tag{30}$$

where the first is by Proposition 1, the second is by the fact that $f^*$ is optimal, and the last is by the property of KL divergence that it is minimized if and only if the distributions are equal. □

# C   Additional Background

This section contains background information on standard concepts that are not directly required to understand our work, but may be useful if the reader wants a more in-depth comparison.

## C.1   Background: Expected Calibration Error for Soft Confidence Evaluation

Our metric missing value calibration error (Section 6.1) is heavily based upon expected calibration error (ECE). While we describe the basics of ECE in terms of how MVCE differs, this section describes it more fully in terms of our overall methodology. The basic intuition of soft confidence values is that if the confidence is $c_{\mathrm{soft}}$ percent, then the classifier should be correct $c_{\mathrm{soft}}$ percent of the time. Ideally, a calibration metric would compare the confidence value to the empirical accuracy conditioned on a specific input. However, given that almost every input is unique, the accuracy is impossible to estimate accurately. Thus, ECE bins samples based on their predicted confidence values and then estimates the empirical accuracy within each bin. The final ECE score is the average over the difference of accuracy and average confidence in each bin. Formally, given a test dataset $\mathcal{D} \equiv \{(x_i, y_i)\}_{i=1}^n$ and the computed the soft predictions $\hat{y}_{\mathrm{soft},i}$ and soft confidence values $\hat{c}_{\mathrm{soft},i}$ for each sample, ECE first partitions the dataset into bins $\mathcal{B}$ based on the soft confidence values $\hat{c}_{\mathrm{soft},i}$. Then, the ECE is defined as: $\mathrm{ECE} = \sum_{B \in \mathcal{B}} \frac{|B|}{|\mathcal{D}|} |\mathrm{acc}(B) - \bar{c}_{\mathrm{soft}}(B)|$, where $\mathrm{acc}(B) \triangleq \frac{1}{|B|} \sum_{i \in B} \mathbb{1}(\hat{y}_{\mathrm{soft},i} = y_i)$ is the empirical classification accuracy and $\bar{c}_{\mathrm{soft}}(B) \triangleq \frac{1}{|B|} \sum_{i \in B} \hat{c}_{\mathrm{soft},i}$ is the average soft confidence. The fact that ECE compares the average confidence to accuracy is related to the fact that the soft decision rule and corresponding confidences are based on the Bayes optimal decision rule, which provides the best accuracy among all possible classifiers.

# D   Extension of Bi-Level DMV Problem with Regularization

While in general the two stage approach is simple and elegant, the learned model might not satisfy a natural constraint that the uncertainty model should converge to a Dirac delta if given fully observed features, i.e., if $\mathcal{O} = \mathcal{F}$, then $p_\theta(\Phi|X_\mathcal{O} = X)$ should converge to a Dirac delta at $\hat{\pi}_\theta(X)$. To enforce this natural constraint, we can ensure that the mean of the distribution is equal to $\hat{\pi}_\theta(X)$ and that the entropy of the distribution is minimized. Specifically, let us define the following loss:

$$\ell_{\mathrm{reg}}(x, \psi, \theta) := \left( \ell_{\mathrm{KL}}(\hat{\pi}_\theta(x), \mathbb{E}_{p_\psi}[\Phi|X_\mathcal{O} = x]) + H(p_\psi(\Phi|X_\mathcal{O} = x)) \right) \tag{31}$$

where $x$ is a complete feature instance, $\ell_{\mathrm{KL}}(p, q)$ is the KL divergence between two probability vectors $p$ and $q$, and $H$ is the standard entropy formulation. Unlike the uncertainty estimation term, this objective function

does not depend on a fixed $\hat{\pi}_\theta$ and thus can be added to both the upper and lower optimization problems:

$$\min_{\psi,\theta} \; \mathbb{E}_{p(X,Y)} [\ell_{\mathrm{CE}}(\hat{\pi}_\theta(X), Y) + \lambda \ell_{\mathrm{reg}}(X, \psi, \theta)] \tag{32}$$

$$\text{s.t. } \psi \in \arg\min_{\tilde{\psi}} \left( \mathbb{E}_{p(X_\mathcal{O}, X_\mathcal{M})} [-\log \hat{p}_{\tilde{\psi}}(\hat{\pi}_\theta(X_\mathcal{O}, X_\mathcal{M}) \mid X_\mathcal{O})] + \lambda \ell_{\mathrm{reg}}(X, \psi, \theta)] \right).$$

Unlike the previous bi-level problem, this problem does not decompose and thus must use more advanced bi-level optimization strategies such as alternating optimization.

## E    Additional Experimental Results

For real world each experiment, we reported two values: MVCE and consistency, as described in Section 3. We used a simple zero-one cost function for estimating confidence, with different mutators based on the dataset. All experiments were run in Ubuntu servers using NVIDIA RTX A5000 GPUs.

**MVCE**  MVCE is our primary metric for comparing methods, reporting how closely the estimate of confidence matches to the likelihood the prediction will change as more information is revealed. Lower MVCE indicates the method is well calibrated and thus gives good confidence estimates. In our experiments, we computed MVCE using 10 bins. When computing MVCE we ran 4 trials of each experiment (or 10 for synthetic) and reported the average of the metric and its standard deviation in order to minimize sources of randomness in the experiment.

**Classifiers**  For all non-synthetic datasets, we used a modified ResNet18 for predictions, replacing the final layer with an appropriately sized layer for the number of classes. For the Direct Missing Value model and the robust to missing values classifier, we additionally replaced the first layer to take 4 channels as an input instead of 3. DMV additionally changed the final activation function from sigmoid (2 classes) or softmax (3 or more classes) to exponential. Both

Table 4: Reported accuracy and expected calibration loss for all models on clean data. Despite the difference in training, neither DMV nor robust to missing classifiers have significantly different accuracy. ECE changed between models, though this difference could likely be mitigated using traditional calibration techniques.

| Dataset | Model | Classes | Accuracy | ECE |
|---|---|---|---|---|
| CelebA - Blond Hair | Classifier | 2 | 93.80% | 0.0397 |
| CelebA - Blond Hair | DMV | 2 | 94.45% | 0.0046 |
| CelebA - Eyeglasses | Classifier | 2 | 99.25% | 0.0053 |
| CelebA - Eyeglasses | DMV | 2 | 99.50% | 0.0035 |
| CelebA - Smiling | Classifier | 2 | 91.85% | 0.0462 |
| CelebA - Smiling | DMV | 2 | 92.85% | 0.0118 |
| MNIST | Classifier | 10 | 99.48% | 0.0009 |
| MNIST | Robust | 10 | 99.59% | 0.0026 |
| MNIST | DMV | 10 | 98.54% | 0.0233 |
| CIFAR10 | Classifier | 10 | 71.77% | 0.0753 |
| CIFAR10 | Robust | 10 | 82.35% | 0.0478 |
| CIFAR10 | DMV | 10 | 79.99% | 0.0561 |
| StarCraftCIFAR10 | Classifier | 10 | 79.90% | 0.0312 |
| StarCraftCIFAR10 | Robust | 10 | 71.84% | 0.0830 |
| StarCraftCIFAR10 | DMV | 10 | 79.78% | 0.0297 |

the standard and robust to missing values classifier were trained using cross entropy loss. We used the SVG optimizer for the standard classifier and Adam for the robust to missing values classifier. Training bash files can be found in the in the codebase for more details.

**Hyperparameters**

For full details on experiment hyperparameters, we have included `.json` files containing the arguments used to our training and evaluation scripts (including random seeds). See the `README` for details on locating them, and the `scripts` folder for example bash files to run the scripts.

**Experiment Setup**

We used a random crop from the larger image size to 224x244 for our model inputs at training time, and a center crop to 224x244 at testing time. For the DMV and the robust to missing values classifier, we replaced the first layer of PyTorch's pretrained ResNet18. Additionally, we replaced the final layer to adjust the class size. All other layers had weights copied over from the pretrained weights found in PyTorch. For our mutator used in both training the DMV and robust classifier and evaluating MVCE, we divided the image into a 4x4 grid of 56x56 pixel regions and gave each region a 50% chance to be removed each time a sample was fetched.

For all three datasets, we cached the average image to use for mean imputation. Both mean imputation and the robust to missing values classifier used three different approaches to estimating variance: 0 variance (a single point prediction), taking half of the maximum possible variance for a Dirichlet distribution, and scaling the predicted probabilities by a factor of 10 (which is multiplied by the calibration constant).

### E.1 CelebA

We made use of the CelebA-HQ dataset (Karras et al., 2017), which contains 10,000 64x64 color images of celebrity faces. We choose this dataset as it was easy to form intuitions about the relationship between missing masks and the CelebA features. Additionally, it was easy to locate pre-trained diffusion models for the dataset. Since CelebA-HQ was designed for training generative models, it lacks target labels, so we obtained the target label information from (Lee et al., 2019). Experiments on the CelebA dataset were run on three different target features: Blond Hair, Eyeglasses, and Smiling. Tables 5, 6, and 7 show CelebA results on the blond hair feature. Tables 8, 9, and 10 show CelebA results on the eyeglasses feature. Tables 11, 12, and 13 show CelebA results on the smiling feature.

**Experiment Setup** An independant ResNet18 classifier with pre-trained initial weights was fine-tuned to predict each feature, making the experiments on CelebA all two class: either the feature is present or absent. We did not train the classifier with robustness to missing values. We used the 64x64 pixel versions of the CelebA images as inputs with no rescaling during preprocessing to better match the output resolution of the pretrained diffusion model; taking advantage of the fact ResNet18 allows variable sized inputs. The DMV model was a similarly constructed ResNet18 fine-tuned with a mutator that randomly selected a missing mask between fully observed, fully missing, top half missing, and bottom half missing. At test time, we used a single mask for the entire experiment, either top half missing or bottom half missing. In the tables below, the words "top" or "bottom" always refer to the half missing.

**Diffusion** For the CelebA dataset, we could take advantage of a pretrained diffusion model to perform experiments. This pretrained diffusion model allowed us to use the Monte Carlo approximation for Missing Value Uncertainty estimation. We could not do the same on other datasets due to the lack of a diffusion model, choosing to forego training more models after realizing the diffusion model approaches are too slow in practice to use. Instead, they serve as a baseline to demonstrate whether DMV is a viable approach.

**Sample Cache** To reduce the time it takes to run multiple experiments, we cached 30 samples from the diffusion model for each mask on each of the 2000 test samples, along with each of the 2000 validation samples for the sake of calibration. This allowed the time to run each experiment with the diffusion model to be relatively close to that of the non-diffusion approaches at the cost of requiring several months to pre-generate all the samples. The single sample times reported in Table 1 for any diffusion approaches takes the time to compute that number of samples from the cache generation and adds it to the time to run that particular method. This caching approach is likely not representative of how the model would be used when deployed, but we do not believe it had any significant impact on the results beyond a small reduction of randomness when running multiple trials. Leveraging this cache limited us to just the two masks in our experiments, though this is not a practical limitation to either method as both DMV and the diffusion model can handle any arbitrary missing feature with the right training.

**Calibration** We calibrated models by making use of the validation dataset split, ensuring that test data remains unseen. Evaluation of calibration is done on the same test data as the original evaluation of MVCE. The calibration constants we found from validation data are reported in tables below under the "scale" heading.

Table 5: Blond Hair is a feature that is often visible in both halves of the image, though may sometimes only be visible in the top half. This causes a smaller average accuracy drop across the baselines. Diffusion and DMV both give the best estimate of the confidence for that drop. Note that the scale column indicates the calibration value used for calibrated MVCE; a value of 1 would indicate the original calibration performed the best.

| CelebA Blond Hair - Mask Average | Acc. Clean | Acc. With Missing Values | Uncalibrated MVCE | Scale | MV Calibrated MVCE | |
|---|---|---|---|---|---|---|
| Diffusion - 1 Sample, 0 Variance | 93.8% | 92.1% (SD 0.0000 ) | 0.0522 (SD 0.0000 ) | - | - | - |
| Diffusion - 1 Sample, 0.5 Max Variance | 93.8% | 92.1% (SD 0.0000 ) | 0.0328 (SD 0.0005 ) | 0.1 | 0.0296 (SD 0.0008 ) | -9.7% |
| Diffusion - 1 Sample, Scale Probability | 93.8% | 92.1% (SD 0.0000 ) | 0.0448 (SD 0.0001 ) | 0.1 | 0.0326 (SD 0.0005 ) | -27.2% |
| Diffusion - 3 Samples - Empirical Variance | 93.8% | 92.4% (SD 0.0000 ) | 0.0205 (SD 0.0003 ) | 0.25 | 0.0176 (SD 0.0008 ) | -14.1% |
| Diffusion - 30 Samples - Empirical Variance | 93.8% | **93.2%** (SD 0.0000 ) | **0.0062** (SD 0.0004 ) | 0.5 | **0.0063** (SD 0.0007 ) | 2.0% |
| Mean Imputation, 0 Variance | 93.8% | 89.4% (SD 0.0000 ) | 0.0775 (SD 0.0000 ) | - | - | - |
| Mean Imputation, 0.5 Max Variance | 93.8% | 89.4% (SD 0.0000 ) | 0.0581 (SD 0.0009 ) | 0.1 | 0.0548 (SD 0.0006 ) | -5.7% |
| Mean Imputation, Scale Probability | 93.8% | 89.4% (SD 0.0000 ) | 0.0704 (SD 0.0003 ) | 0.1 | 0.0583 (SD 0.0004 ) | -17.2% |
| Direct Missing Value (DMV) | 94.5% | **93.0%** (SD 0.0000 ) | **0.0286** (SD 0.0002 ) | 7.5 | **0.0059** (SD 0.0011 ) | -79.2% |

Table 6: While blond hair is typically visible in both halves, it is more likely to be in the top half, causing a small accuracy drop for the imputation methods. This is less notable than the smiling method, and does not impact the ranking of methods.

| CelebA Blond Hair - Top Missing | Acc. Clean | Acc. With Missing Values | Uncalibrated MVCE | Scale | MV Calibrated MVCE | |
|---|---|---|---|---|---|---|
| Diffusion - 1 Sample, 0 Variance | 93.8% | 90.8% (SD 0.0000 ) | 0.0665 (SD 0.0000 ) | - | - | - |
| Diffusion - 1 Sample, 0.5 Max Variance | 93.8% | 90.8% (SD 0.0000 ) | 0.0472 (SD 0.0001 ) | 0.1 | 0.0443 (SD 0.0001 ) | -6.1% |
| Diffusion - 1 Sample, Scale Probability | 93.8% | 90.8% (SD 0.0000 ) | 0.0599 (SD 0.0000 ) | 0.1 | 0.0473 (SD 0.0001 ) | -21.1% |
| Diffusion - 3 Samples - Empirical Variance | 93.8% | 91.0% (SD 0.0000 ) | 0.0253 (SD 0.0001 ) | 0.25 | 0.0224 (SD 0.0009 ) | -11.6% |
| Diffusion - 30 Samples - Empirical Variance | 93.8% | **92.5%** (SD 0.0000 ) | **0.0083** (SD 0.0003 ) | 0.5 | **0.0073** (SD 0.0008 ) | -12.0% |
| Mean Imputation, 0 Variance | 93.8% | 85.7% (SD 0.0000 ) | 0.1005 (SD 0.0000 ) | - | - | - |
| Mean Imputation, 0.5 Max Variance | 93.8% | 85.7% (SD 0.0000 ) | 0.0901 (SD 0.0010 ) | 0.1 | 0.0880 (SD 0.0001 ) | -2.4% |
| Mean Imputation, Scale Probability | 93.8% | 85.7% (SD 0.0000 ) | 0.0978 (SD 0.0001 ) | 0.1 | 0.0901 (SD 0.0003 ) | -7.9% |
| Direct Missing Value (DMV) | 94.5% | **92.0%** (SD 0.0000 ) | **0.0323** (SD 0.0002 ) | 7.5 | **0.0058** (SD 0.0017 ) | -82.0% |

Table 7: Blond hair is almost always visible in the top half, making the accuracy remain high even among the baselines. Despite this, they often still get low MVCE, in this case due to under-estimating the confidence.

| CelebA Blond Hair - Bottom Missing | Acc. Clean | Acc. With Missing Values | Uncalibrated MVCE | Scale | MV Calibrated MVCE | |
|---|---|---|---|---|---|---|
| Diffusion - 1 Sample, 0 Variance | 93.8% | 93.5% (SD 0.0000 ) | 0.0380 (SD 0.0000 ) | - | - | - |
| Diffusion - 1 Sample, 0.5 Max Variance | 93.8% | 93.5% (SD 0.0000 ) | 0.0185 (SD 0.0007 ) | 0.1 | 0.0150 (SD 0.0012 ) | -18.8% |
| Diffusion - 1 Sample, Scale Probability | 93.8% | 93.5% (SD 0.0000 ) | 0.0297 (SD 0.0002 ) | 0.1 | 0.0179 (SD 0.0007 ) | -39.7% |
| Diffusion - 3 Samples - Empirical Variance | 93.8% | **93.8%** (SD 0.0000 ) | 0.0156 (SD 0.0004 ) | 0.25 | 0.0127 (SD 0.0009 ) | -18.2% |
| Diffusion - 30 Samples - Empirical Variance | 93.8% | 93.8% (SD 0.0000 ) | **0.0040** (SD 0.0005 ) | 0.5 | **0.0053** (SD 0.0006 ) | 30.9% |
| Mean Imputation, 0 Variance | 93.8% | 93.0% (SD 0.0000 ) | 0.0545 (SD 0.0000 ) | - | - | - |
| Mean Imputation, 0.5 Max Variance | 93.8% | 93.0% (SD 0.0000 ) | 0.0261 (SD 0.0009 ) | 0.1 | 0.0216 (SD 0.0010 ) | -17.2% |
| Mean Imputation, Scale Probability | 93.8% | 93.0% (SD 0.0000 ) | 0.0430 (SD 0.0004 ) | 0.1 | 0.0265 (SD 0.0005 ) | -38.4% |
| Direct Missing Value (DMV) | 94.5% | **94.0%** (SD 0.0000 ) | **0.0249** (SD 0.0002 ) | 7.5 | **0.0060** (SD 0.0005 ) | -75.7% |

Table 8: Eyeglasses are often visible in both halves of the image, causing a minimal drop in accuracy regardless of mask. This leads to MVCE primarily penalizing underestimated confidence.

| CelebA Eyeglasses - Mask Average | Acc. Clean | Acc. With Missing Values | Uncalibrated MVCE | MV Calibrated Scale | MV Calibrated MVCE | |
|---|---|---|---|---|---|---|
| Diffusion - 1 Sample, 0 Variance | 99.3% | 99.0% (SD 0.0000 ) | 0.0058 (SD 0.0000 ) | - | - | - |
| Diffusion - 1 Sample, 0.5 Max Variance | 99.3% | 99.0% (SD 0.0000 ) | 0.0045 (SD 0.0002 ) | 0.1 | 0.0040 (SD 0.0001 ) | -10.9% |
| Diffusion - 1 Sample, Scale Probability | 99.3% | 99.0% (SD 0.0000 ) | 0.0058 (SD 0.0001 ) | 0.1 | 0.0045 (SD 0.0002 ) | -22.4% |
| Diffusion - 3 Samples - Empirical Variance | 99.3% | **99.0%** (SD 0.0000 ) | 0.0037 (SD 0.0002 ) | 0.5 | 0.0037 (SD 0.0001 ) | -1.1% |
| Diffusion - 30 Samples - Empirical Variance | 99.3% | **99.0%** (SD 0.0000 ) | **0.0021** (SD 0.0001 ) | 7.5 | **0.0033** (SD 0.0003 ) | 54.4% |
| Mean Imputation, 0 Variance | 99.3% | 97.5% (SD 0.0000 ) | 0.0220 (SD 0.0000 ) | - | - | - |
| Mean Imputation, 0.5 Max Variance | 99.3% | 97.5% (SD 0.0000 ) | 0.0111 (SD 0.0003 ) | 0.75 | 0.0109 (SD 0.0004 ) | -1.7% |
| Mean Imputation, Scale Probability | 99.3% | 97.5% (SD 0.0000 ) | 0.0192 (SD 0.0003 ) | 0.1 | 0.0113 (SD 0.0004 ) | -41.0% |
| Direct Missing Value (DMV) | 99.5% | **99.1%** (SD 0.0000 ) | **0.0029** (SD 0.0001 ) | 5 | **0.0036** (SD 0.0001 ) | 23.4% |

Table 9: While intuitively eyeglasses are a feature in the top half, the edges are often on the boundary allowing high accuracy from just the bottom half as well.

| CelebA Eyeglasses - Top Missing | Acc. Clean | Acc. With Missing Values | Uncalibrated MVCE | MV Calibrated Scale | MV Calibrated MVCE | |
|---|---|---|---|---|---|---|
| Diffusion - 1 Sample, 0 Variance | 99.3% | **99.1%** (SD 0.0000 ) | 0.0055 (SD 0.0000 ) | - | - | - |
| Diffusion - 1 Sample, 0.5 Max Variance | 99.3% | **99.1%** (SD 0.0000 ) | 0.0041 (SD 0.0002 ) | 0.1 | 0.0033 (SD 0.0000 ) | -19.5% |
| Diffusion - 1 Sample, Scale Probability | 99.3% | **99.1%** (SD 0.0000 ) | 0.0053 (SD 0.0001 ) | 0.1 | 0.0040 (SD 0.0003 ) | -25.8% |
| Diffusion - 3 Samples - Empirical Variance | 99.3% | 99.0% (SD 0.0000 ) | 0.0034 (SD 0.0002 ) | 0.5 | **0.0031** (SD 0.0001 ) | -9.4% |
| Diffusion - 30 Samples - Empirical Variance | 99.3% | 99.0% (SD 0.0000 ) | **0.0024** (SD 0.0002 ) | 7.5 | 0.0042 (SD 0.0004 ) | 75.1% |
| Mean Imputation, 0 Variance | 99.3% | 97.6% (SD 0.0000 ) | 0.0230 (SD 0.0000 ) | - | - | - |
| Mean Imputation, 0.5 Max Variance | 99.3% | 97.6% (SD 0.0000 ) | 0.0062 (SD 0.0003 ) | 0.75 | 0.0061 (SD 0.0006 ) | -1.5% |
| Mean Imputation, Scale Probability | 99.3% | 97.6% (SD 0.0000 ) | 0.0182 (SD 0.0000 ) | 0.1 | 0.0067 (SD 0.0005 ) | -63.4% |
| Direct Missing Value (DMV) | 99.5% | **99.0%** (SD 0.0000 ) | **0.0025** (SD 0.0001 ) | 5 | **0.0026** (SD 0.0001 ) | 4.8% |

Table 10: Eyeglasses are clearly visible in the top half of the image, allowing high accuracy regardless of method of handling missing values. The direct methods manage to minimize MVCE by not underestimating confidence.

| CelebA Eyeglasses - Bottom Missing | Acc. Clean | Acc. With Missing Values | Uncalibrated MVCE | MV Calibrated Scale | MV Calibrated MVCE | |
|---|---|---|---|---|---|---|
| Diffusion - 1 Sample, 0 Variance | 99.3% | 98.9% (SD 0.0000 ) | 0.0060 (SD 0.0000 ) | - | - | - |
| Diffusion - 1 Sample, 0.5 Max Variance | 99.3% | 98.9% (SD 0.0000 ) | 0.0050 (SD 0.0002 ) | 0.1 | 0.0048 (SD 0.0001 ) | -3.9% |
| Diffusion - 1 Sample, Scale Probability | 99.3% | 98.9% (SD 0.0000 ) | 0.0062 (SD 0.0000 ) | 0.1 | 0.0050 (SD 0.0000 ) | -19.5% |
| Diffusion - 3 Samples - Empirical Variance | 99.3% | **99.0%** (SD 0.0000 ) | 0.0040 (SD 0.0002 ) | 0.5 | 0.0043 (SD 0.0002 ) | 5.9% |
| Diffusion - 30 Samples - Empirical Variance | 99.3% | **99.0%** (SD 0.0000 ) | **0.0018** (SD 0.0001 ) | 7.5 | **0.0023** (SD 0.0001 ) | 26.9% |
| Mean Imputation, 0 Variance | 99.3% | 97.5% (SD 0.0000 ) | 0.0210 (SD 0.0000 ) | - | - | - |
| Mean Imputation, 0.5 Max Variance | 99.3% | 97.5% (SD 0.0000 ) | 0.0160 (SD 0.0003 ) | 0.75 | 0.0157 (SD 0.0002 ) | -1.8% |
| Mean Imputation, Scale Probability | 99.3% | 97.5% (SD 0.0000 ) | 0.0202 (SD 0.0004 ) | 0.1 | 0.0160 (SD 0.0003 ) | -20.8% |
| Direct Missing Value (DMV) | 99.5% | **99.1%** (SD 0.0000 ) | **0.0033** (SD 0.0001 ) | 5 | **0.0045** (SD 0.0001 ) | 37.3% |

Table 11: The smiling feature has the worst performance of DMV out of all three features, though it is still on average better than the baselines. This suggests a need for further optimization of the model.

| CelebA Smiling - Mask Average | Acc. Clean | Acc. With Missing Values | Uncalibrated MVCE | MV Calibrated Scale | MV Calibrated MVCE | |
|---|---|---|---|---|---|---|
| Diffusion - 1 Sample, 0 Variance | 91.9% | 84.1% (SD 0.0000 ) | 0.1357 (SD 0.0000 ) | - | - | - |
| Diffusion - 1 Sample, 0.5 Max Variance | 91.9% | 84.1% (SD 0.0000 ) | 0.1028 (SD 0.0015 ) | 0.1 | 0.0990 (SD 0.0007 ) | -3.7% |
| Diffusion - 1 Sample, Scale Probability | 91.9% | 84.1% (SD 0.0000 ) | 0.1226 (SD 0.0002 ) | 0.1 | 0.1028 (SD 0.0005 ) | -16.2% |
| Diffusion - 3 Samples - Empirical Variance | 91.9% | 85.6% (SD 0.0000 ) | 0.0555 (SD 0.0008 ) | 0.25 | 0.0530 (SD 0.0006 ) | -4.6% |
| Diffusion - 30 Samples - Empirical Variance | 91.9% | **86.4%** (SD 0.0000 ) | **0.0163** (SD 0.0018 ) | 0.5 | **0.0144** (SD 0.0028 ) | -11.5% |
| Mean Imputation, 0 Variance | 91.9% | 70.5% (SD 0.0000 ) | 0.2633 (SD 0.0000 ) | - | - | - |
| Mean Imputation, 0.5 Max Variance | 91.9% | 70.5% (SD 0.0000 ) | 0.2335 (SD 0.0007 ) | 0.1 | 0.2281 (SD 0.0002 ) | -2.3% |
| Mean Imputation, Scale Probability | 91.9% | 70.5% (SD 0.0000 ) | 0.2548 (SD 0.0003 ) | 0.1 | 0.2335 (SD 0.0006 ) | -8.4% |
| Direct Missing Value (DMV) | 92.8% | **77.1%** (SD 0.0000 ) | **0.0711** (SD 0.0004 ) | 2.5 | **0.0695** (SD 0.0010 ) | -2.2% |

Table 12: Since smiling is primarily in the bottom half, the top half missing does not significantly impact the prediction leading to a high accuracy prediction. Interestingly, the simple heuristic of half the maximum variance manages to minimize MVCE. For Monte Carlo, this suggests too much of the top half is used in the prediction, while DMV may simply estimate too little confidence overall.

| CelebA Smiling - Top Missing | Acc. Clean | Acc. With Missing Values | Uncalibrated MVCE | MV Calibrated Scale | MV Calibrated MVCE | |
|---|---|---|---|---|---|---|
| Diffusion - 1 Sample, 0 Variance | 91.9% | 91.0% (SD 0.0000 ) | 0.0390 (SD 0.0000 ) | - | - | - |
| Diffusion - 1 Sample, 0.5 Max Variance | 91.9% | 91.0% (SD 0.0000 ) | **0.0096** (SD 0.0023 ) | 0.1 | **0.0072** (SD 0.0011 ) | -24.8% |
| Diffusion - 1 Sample, Scale Probability | 91.9% | 91.0% (SD 0.0000 ) | 0.0265 (SD 0.0003 ) | 0.1 | 0.0095 (SD 0.0008 ) | -64.2% |
| Diffusion - 3 Samples - Empirical Variance | 91.9% | 91.8% (SD 0.0000 ) | 0.0152 (SD 0.0005 ) | 0.25 | 0.0090 (SD 0.0005 ) | -40.6% |
| Diffusion - 30 Samples - Empirical Variance | 91.9% | **91.8%** (SD 0.0000 ) | 0.0126 (SD 0.0004 ) | 0.5 | 0.0088 (SD 0.0011 ) | -30.0% |
| Mean Imputation, 0 Variance | 91.9% | 91.1% (SD 0.0000 ) | 0.0450 (SD 0.0000 ) | - | - | - |
| Mean Imputation, 0.5 Max Variance | 91.9% | 91.1% (SD 0.0000 ) | **0.0218** (SD 0.0002 ) | 0.1 | **0.0173** (SD 0.0001 ) | -20.6% |
| Mean Imputation, Scale Probability | 91.9% | 91.1% (SD 0.0000 ) | 0.0382 (SD 0.0002 ) | 0.1 | 0.0218 (SD 0.0004 ) | -43.0% |
| Direct Missing Value (DMV) | 92.8% | **92.7%** (SD 0.0000 ) | 0.0363 (SD 0.0002 ) | 2.5 | 0.0277 (SD 0.0002 ) | -23.6% |

Table 13: Without the bottom half, the main indicator of smiling is missing, leading to a significant accuracy drop among baselines. DMV in this case is not ideally optimized, making it barely better than imputation.

| CelebA Smiling - Bottom Missing | Acc. Clean | Acc. With Missing Values | Uncalibrated MVCE | MV Calibrated Scale | MV Calibrated MVCE | |
|---|---|---|---|---|---|---|
| Diffusion - 1 Sample, 0 Variance | 91.9% | 77.2% (SD 0.0000 ) | 0.2325 (SD 0.0000 ) | - | - | - |
| Diffusion - 1 Sample, 0.5 Max Variance | 91.9% | 77.2% (SD 0.0000 ) | 0.1960 (SD 0.0002 ) | 0.1 | 0.1908 (SD 0.0001 ) | -2.7% |
| Diffusion - 1 Sample, Scale Probability | 91.9% | 77.2% (SD 0.0000 ) | 0.2188 (SD 0.0001 ) | 0.1 | 0.1960 (SD 0.0002 ) | -10.4% |
| Diffusion - 3 Samples - Empirical Variance | 91.9% | 79.5% (SD 0.0000 ) | 0.0959 (SD 0.0011 ) | 0.25 | 0.0969 (SD 0.0007 ) | 1.1% |
| Diffusion - 30 Samples - Empirical Variance | 91.9% | **81.0%** (SD 0.0000 ) | **0.0199** (SD 0.0028 ) | 0.5 | **0.0199** (SD 0.0042 ) | 0.2% |
| Mean Imputation, 0 Variance | 91.9% | 49.9% (SD 0.0000 ) | 0.4815 (SD 0.0000 ) | - | - | - |
| Mean Imputation, 0.5 Max Variance | 91.9% | 49.9% (SD 0.0000 ) | 0.4453 (SD 0.0010 ) | 0.1 | 0.4390 (SD 0.0004 ) | -1.4% |
| Mean Imputation, Scale Probability | 91.9% | 49.9% (SD 0.0000 ) | 0.4714 (SD 0.0004 ) | 0.1 | 0.4453 (SD 0.0009 ) | -5.5% |
| Direct Missing Value (DMV) | 92.8% | **61.4%** (SD 0.0000 ) | **0.1058** (SD 0.0006 ) | 2.5 | **0.1113** (SD 0.0016 ) | 5.2% |

### E.2 Multiclass Datasets

**MNIST**

For a simple baseline, we used MNIST (Deng, 2012), which consists of hand-drawn black and white 28x28 digits between 0 and 9. Class labels are easy to interpret, and the single channel makes it easier to learn the models, though it is also a less challenging problem under missing values. Results for the experiments on MNIST are shown in Table 14.

Table 14: MNIST was an easier dataset to predict with missing values, leading to few cases where more information was needed to make good predictions (and thus most models were under-confident). Despite this, DMV was still comparable, and could be calibrated to perform nearly the same as the robust classifier. While it was possible to similarly calibrate the robust classifier with uncertainty heuristics, the under confident predictions reduced the uncertainty significantly. Overall, this dataset ended up as too easy of a task for traditional neural network to handle so its harder to see the benefit from MVU.

| MNIST Method \ Missing | Acc. Clean | Acc. With Missing Values | Uncalibrated MVCE | MV Calibrated Scale | MVCE | |
|---|---|---|---|---|---|---|
| Mean Imputation, 0 Variance | 99.5% | 58.7% (SD 0.0024 ) | 0.4116 (SD 0.0027 ) | - | - | - |
| Mean Imputation, 0.5 Max Variance | 99.5% | 58.6% (SD 0.0022 ) | 0.1900 (SD 0.0025 ) | 0.1 | 0.1644 (SD 0.0022 ) | -14% |
| Mean Imputation, Scale Probability | 99.5% | 59.0% (SD 0.0021 ) | 0.3053 (SD 0.0013 ) | 0.1 | 0.1903 (SD 0.0044 ) | -38% |
| Missing Robust, 0 Variance | 99.6% | 96.9% (SD 0.0017 ) | 0.0296 (SD 0.0017 ) | - | - | - |
| Missing Robust, 0.5 Max Variance | 99.6% | **96.9%** (SD 0.0005 ) | **0.0031** (SD 0.0006 ) | 1.0 | **0.0031** (SD 0.0006 ) | 0% |
| Missing Robust, Scale Probability | 99.6% | 96.9% (SD 0.0004 ) | 0.0168 (SD 0.0005 ) | 0.1 | 0.0048 (SD 0.0004 ) | -71% |
| Direct Missing Value (DMV) | 98.5% | 92.7% (SD 0.0016 ) | 0.0736 (SD 0.0027 ) | 10.0 | 0.0354 (SD 0.0010 ) | -52% |

**CIFAR10** CIFAR10 (Krizhevsky et al., 2009) is a well known baseline in Machine Learning, containing 60,000 32x32 color images of airplanes, automobiles, birds, cats, deer, dogs, frogs, horses, ships, and trucks. Like CelebA, the class labels are easy to interpret, though there is typically a much less obvious relationship between particular regions in the image and the prediction making CIFAR10 a good intermediate difficulty experiment. Additionally, the multi-class setup provides some difficulties that were not seen in single class. Results for the experiments on CIFAR10 are shown in Table 15.

Table 15: The DMV method and the robust classifier both have comparable mutated accuracy, though DMV notably produces better confidence estimates as its able to change confidence with respect to each sample. Reducing MVCE on the robust classifier through uncertainty heuristics leads notably better MVCE, but it still did not compare. Overall, this dataset shows the value of DMV for estimating uncertainty that indicates the prediction likely shifted.

| CIFAR10 Method \ Missing | Acc. Clean | Acc. With Missing Values | Uncalibrated MVCE | MV Calibrated Scale | MVCE | |
|---|---|---|---|---|---|---|
| Mean Imputation, 0 Variance | 71.8% | 32.1% (SD 0.0055 ) | 0.6471 (SD 0.0030 ) | - | - | - |
| Mean Imputation, 0.5 Max Variance | 71.8% | 32.4% (SD 0.0050 ) | 0.3092 (SD 0.0056 ) | 0.1 | 0.2855 (SD 0.0030 ) | -8% |
| Mean Imputation, Scale Probability | 71.8% | 32.0% (SD 0.0027 ) | 0.4776 (SD 0.0026 ) | 0.1 | 0.3115 (SD 0.0033 ) | -35% |
| Missing Robust, 0 Variance | 82.3% | **74.7%** (SD 0.0050 ) | 0.1604 (SD 0.0033 ) | - | - | - |
| Missing Robust, 0.5 Max Variance | 82.3% | 74.7% (SD 0.0022 ) | 0.0804 (SD 0.0041 ) | 0.1 | 0.0740 (SD 0.0009 ) | -8% |
| Missing Robust, Scale Probability | 82.3% | 74.7% (SD 0.0014 ) | 0.1349 (SD 0.0010 ) | 0.1 | 0.0807 (SD 0.0017 ) | -40% |
| Direct Missing Value (DMV) | 80.0% | 72.3% (SD 0.0011 ) | **0.0455** (SD 0.0050 ) | 1.0 | **0.0455** (SD 0.0050 ) | 0% |

**StarCraftCIFAR10**

StarCraftCIFAR10 (Kulinski et al., 2023) is a dataset meant to simulate a battlefield scenario with data created from replays of the game StarCraft II. The dataset follows the same format as CIFAR10, though the classes are replaced with 5 maps and a time of game (either beginning or end). The map part of the class is easily interpretable for those familiar with the game, while the time of game tends to be more difficult for

a human to identify making this a more difficult dataset. It was chosen partly for the similarity in format to CIFAR10, and partly to expand upon our sensor network motivation. Results for the experiments on StarCraftCIFAR10 are shown in Table 16.

Table 16: The map prediction for StarCraftCIFAR10 makes it easier to achieve comparable accuracy on mutated data compared to clean data even using simple heuristics. The time of game prediction however is more difficult than either of the prior prediction tasks, which requires a wholistic approach to missing values to fully handle well, hence accuracy being lower across the board compared to other datasets.

| StarCraft Method \ Missing | Acc. Clean | Acc. With Missing Values | Uncalibrated MVCE | MV Calibrated Scale | MV Calibrated MVCE | |
|---|---|---|---|---|---|---|
| Mean Imputation, 0 Variance | 79.9% | 70.5% (SD 0.0039 ) | 0.2231 (SD 0.0035 ) | - | - | - |
| Mean Imputation, 0.5 Max Variance | 79.9% | 70.6% (SD 0.0029 ) | 0.0493 (SD 0.0046 ) | 2.5 | 0.0214 (SD 0.0026 ) | -57% |
| Mean Imputation, Scale Probability | 79.9% | 70.6% (SD 0.0078 ) | 0.0989 (SD 0.0060 ) | 0.25 | 0.0233 (SD 0.0044 ) | -76% |
| Missing Robust, 0 Variance | 71.8% | 71.3% (SD 0.0031 ) | 0.1278 (SD 0.0032 ) | - | - | - |
| Missing Robust, 0.5 Max Variance | 71.8% | 71.2% (SD 0.0032 ) | 0.0809 (SD 0.0009 ) | 2.5 | 0.0582 (SD 0.0032 ) | -28% |
| Missing Robust, Scale Probability | 71.8% | 71.2% (SD 0.0021 ) | 0.0937 (SD 0.0035 ) | 0.25 | **0.0571** (SD 0.0008 ) | -39% |
| Direct Missing Value (DMV) | 79.8% | **78.5%** (SD 0.0026 ) | **0.0201** (SD 0.0016 ) | 0.5 | **0.0113** (SD 0.0016 ) | -44% |

**Experiment Setup**

We fine-tuned a modified pretrained ResNet18 model for both classifiers and DMV for both CIFAR10 and StarCraftCIFAR10. Since MNIST are only a single channel, we modified the ResNet18 model to use a single channel input for the standard classifier, and a 2 channel input for the DMV and robust classifiers. Other datasets used 3 channels for standard classifiers and 4 for the DMV and robust classifier. We rescaled the images to 224x244 during preprocessing to better match the expected input size for ResNet18. For our mutator used in both training the DMV and evaluating MVCE, we divided the image into a 4x4 grid of 56x56 pixel regions and gave each region a 50% chance to be removed each time a sample was fetched. As we lacked a pretrained diffusion model this datasets and it was too slow to use in practice, we skipped all diffusion related methods for experiments on both datasets.

**Calibration**

We calibrated models by making use of the validation dataset split, ensuring that test data remains unseen. Evaluation of calibration is done on the same test data as the original evaluation of MVCE. Since the cost function becomes immensely more complex with more than 2 variables, we simply calibrated the model on a single zero-one cost function. This means CIFAR10 and StarCraftCIFAR10 both likely get slightly better results from calibration as we know the testing environment.

### E.3   Ablation Study on Alternative Mutators

Following the same DMV model setup from subsection E.2, we trained three different DMV models based on three different mutators. We reused the mutator used in the previous experiments which is called MCAR, and created two MNAR mutators. As before, we divided the images into a 4 by 4 grid of patches, such that an entire patch was either present or absent. For the MCAR mutator, each patch had a 50% chance of being dropped as described in Table E. For MNAR, ran the patch contents (which was a 3 dimensional tensor) through a Sigmoid giving the form $chance = (1 + e^{S \cdot A(patch)})^{-1}$ where $A(patch)$ is the aggregator and $S$ is a sharpness parameter.

On the StarcraftCIFAR10 dataset, the patch size was 56x56x2, as we only included the player 1 and player 2 units, not the environment channel. We used a max aggregator that passed the largest value in the patch through the sigmoid, which represents a unit present in either player within that patch (or a unit recently present if no unit was currently present). MNAR (high) used a value of -1.5 for sharpness which causes regions with units to be more likely to drop, while MNAR (low) used 1.5 for sharpness which caused regions without units to be more likely to drop. Table 17 shows the results of this experiment on StarcraftCIFAR10. Like before, we ran 4 trials of each experiment due to the randomness in sampling some of the distributions

such as the mutator, so the table in the appendix includes the standard devitions and averages of the 4 experiments run. Further discussion on the results can be found in section 7.

Table 17: Standard deviations on the MVCE scores for the Starcraft show the rankings of different models are not just a result of random fluctuations. The best average case performance is on the model trained on MCAR, while both MNAR setups performed better when evaluated under the matching mutator, and worse when evaluated under the opposite mutator. This suggests overall that training on non-MCAR data should only be done if the operator is certain that structure matches the expected test time mutator.

| Training Mutator | Clean | Test Time Missingness | | | | | | |
| | | MCAR | | | MNAR (high) | | MNAR (low) | |
| | Acc | Acc | MVCE | | Acc | MVCE | Acc | MVCE |
| MCAR | 79.78% | **79.78%** (SD 0.0026 ) | **0.0201** (SD 0.0016 ) | | 75.09% (SD 0.0024 ) | 0.0863 (SD 0.0020 ) | 78.86% (SD 0.0008 ) | **0.0090** (SD 0.0009 ) |
| MNAR (high) | 80.18% | 76.51% (SD 0.0016 ) | 0.0412 (SD 0.0006 ) | | **76.18%** (SD 0.0019 ) | **0.0529** (SD 0.0022 ) | 76.49% (SD 0.0016 ) | 0.0259 (SD 0.0006 ) |
| MNAR (low) | 80.63% | 79.25% (SD 0.0016 ) | 0.0407 (SD 0.0015 ) | | 71.43% (SD 0.0031 ) | 0.1358 (SD 0.0026 ) | **80.17%** (SD 0.0006 ) | 0.0102 (SD 0.0012 ) |

## E.4 Synthetic

To briefly validate our MVCE metric and our post-hoc calibration, we created a simple Gaussian Distribution with two variables, $X_1$ and $X_2$, using $\mathbb{E}[X] = (0,0)$, $\mathrm{Var}(X_1) = 0.3$, $\mathrm{Var}(X_2) = 1$, and $\mathrm{corr}(X_1, X_2) = 0.7$. This distribution is visualized in Figure 1. From this setup, we generated 10000 samples and treated those as our ground truth testing dataset. In addition, we constructed a simple classifier as $\mathrm{sigmoid}(x_1 + x_2 - 1)$ which was used to generate the label through a Bernoulli distribution, along with using it for making our predictions during the computation of MVU and MVCE.

**Experiment Setup**

The ground-truth MVU can be estimated via Monte Carlo by using the the same model that generated the data as our generator $p(X_{\mathcal{M}}|x_{\mathcal{O}})$. We compare the ground-truth missing conditional distribution with perturbations of this distribution to check that the MVCE increases if the MVU distribution shifts. We mutated the generator by increasing or decreasing the correlation between the two variables, and scaling up or scaling down the entire covariance matrix. In addition, we compared against a single sample imputation, which is comparable to taking a single sample from the diffusion model on the CelebA dataset.

**Calibration** In order to verify post-hoc calibration worked, we first calibrated the synthetic dataset. Like the CelebA method, we calibrated using a range of values. To simulate the testing environment, we calibrated using a second set of 10000 samples as our calibration dataset. For the expectation over cost functions, we made a set of cost functions with 0 cost when $y = a$, $t$ loss when $y = 0, a = 1$, and $1 - t$ cost when $y = 1, a = 0$. To perform the expectation, we created a set of $t$ values from 0.1 to 0.9 in 0.1 increments, and then randomly choose a $t$ value in every batch while computing MVCE.

**Results** Table 18 and Table 19 show results of this experiment. MVCE correctly penalizes generators which changed variance or correlation compared to ground truth. Calibration manages to reduce MVCE for models more distant from ground truth, but is not a substitute for improving the generator.

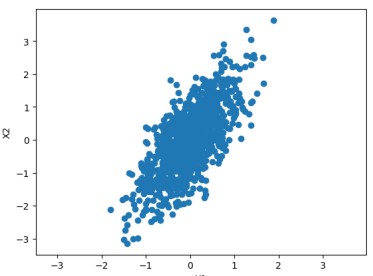

Figure 1: Example distribution of random samples from the synthetic dataset generated to test Missing Value Prediction Uncertainty.

Table 18: Regardless of which feature is missing, we find ground truth overall outperforms the alternatives. The imputation approaches with variance heuristics perform notably poorly. Generator approaches approach the ground truth, but still see a notable increase in MVCE when considering both features, especially before calibration.

| Method \ Missing | Calib. Scale | X1 Uncalibrated MVCE | X1 MV Calibrated MVCE | | X2 Uncalibrated MVCE | X2 MV Calibrated MVCE | |
|---|---|---|---|---|---|---|---|
| 1 Sample Imputation, 0 Variance | - | 0.0983 (SD 0.0018 ) | - | - | 0.1768 (SD 0.0025 ) | - | - |
| 1 Sample Imputation, 0.99 Max Variance | 0.1 | 0.1392 (SD 0.0017 ) | 0.1054 (SD 0.0026 ) | -24% | 0.0616 (SD 0.0033 ) | 0.0628 (SD 0.0018 ) | 2% |
| Mean Imputation, 0 Variance | - | 0.0718 (SD 0.0000 ) | - | - | 0.1235 (SD 0.0000 ) | - | - |
| Mean Imputation, 0.99 Max Variance | 0.1 | 0.1692 (SD 0.0001 ) | 0.1164 (SD 0.0015 ) | -31% | 0.1264 (SD 0.0002 ) | 0.0817 (SD 0.0024 ) | -35% |
| Generator Correlation 0.7 → 0.6 | 1 | 0.0054 (SD 0.0006 ) | 0.0054 (SD 0.0006 ) | 0% | 0.0160 (SD 0.0008 ) | 0.0160 (SD 0.0008 ) | 0% |
| Generator Correlation 0.7 → 0.8 | 0.5 | 0.0167 (SD 0.0006 ) | 0.0058 (SD 0.0012 ) | -65% | 0.0239 (SD 0.0004 ) | 0.0094 (SD 0.0009 ) | -61% |
| Generator Covariance × 0.25 | 0.25 | 0.0372 (SD 0.0007 ) | 0.0057 (SD 0.0004 ) | -85% | 0.0604 (SD 0.0007 ) | 0.0115 (SD 0.0011 ) | -81% |
| Generator Covariance × 4.0 | 5 | 0.0569 (SD 0.0013 ) | 0.0092 (SD 0.0010 ) | -84% | 0.0973 (SD 0.0016 ) | **0.0062** (SD 0.0017 ) | -94% |
| **Ground Truth Generator** | 1 | **0.0051** (SD 0.0007 ) | **0.0051** (SD 0.0007 ) | 0% | **0.0085** (SD 0.0009 ) | 0.0085 (SD 0.0009 ) | 0% |

Table 19: With synthetic data, the ground truth model leads to the lowest MVCE as expected. Mutating the generator or using imputation with a heuristic for estimating MVU both lead to higher MVCE. Post-hoc calibration can greatly reduce MVCE, though it is not a substitute for improving the generator as most of the differences are not a simple scaling.

| Feature Average Method \ Missing | Uncalibrated MVCE | Scale | MV Calibrated MVCE | |
|---|---|---|---|---|
| 1 Sample Imputation, 0 Variance | 0.1376 (SD 0.0021 ) | - | - | - |
| 1 Sample Imputation, 0.99 Max Variance | 0.1004 (SD 0.0025 ) | 0.1 | 0.0841 (SD 0.0022 ) | -16% |
| Mean Imputation, 0 Variance | 0.0976 (SD 0.0000 ) | - | - | - |
| Mean Imputation, 0.99 Max Variance | 0.1478 (SD 0.0002 ) | 0.1 | 0.0991 (SD 0.0019 ) | -33% |
| Generator Correlation 0.7 → 0.6 | 0.0107 (SD 0.0007 ) | 1 | 0.0107 (SD 0.0007 ) | 0% |
| Generator Correlation 0.7 → 0.8 | 0.0203 (SD 0.0005 ) | 0.5 | 0.0076 (SD 0.0010 ) | -63% |
| Generator Covariance × 0.25 | 0.0488 (SD 0.0007 ) | 0.25 | 0.0086 (SD 0.0008 ) | -82% |
| Generator Covariance × 4.0 | 0.0771 (SD 0.0014 ) | 5 | 0.0077 (SD 0.0014 ) | -90% |
| **Ground Truth Generator** | **0.0068** (SD 0.0008 ) | 1 | **0.0068** (SD 0.0008 ) | 0% |

