# OpenReview forum: "Missing Value Uncertainty: Could Collecting Missing Values Change the Prediction?"
_TMLR — Decision pending for TMLR_

### Review · Reviewer_aXJu · 2026-03-29

**Summary Of Contributions:**

This paper introduces a framework for quantifying uncertainty arising from missing input features at inference time. It formalizes Missing Value Uncertainty (MVU) as the distribution over predictions induced by unknown feature values, providing a principled way to characterize how incomplete inputs affect model outputs. The authors further propose using hard voting, rather than soft voting, to assess whether collecting additional features is likely to change the prediction. To efficiently estimate the MVU distribution, the paper introduces the Direct Missing Value (DMV) estimator, which learns a model to approximate $p(\Phi \mid \mathbf{X}_O)$ without relying on costly sampling or imputation. In addition, the authors propose the Missing Value Calibration Error (MVCE) to evaluate the calibration of hard confidence, along with a post-hoc calibration procedure to improve its reliability.

**Additional Comments:**

The authors have made edits that improve the clarity of the work. In summary, the work proposes modeling MVU with another ML model to bypass the need for expensive Monte Carlo sampling when estimating prediction uncertainty caused by missing values. I view this as an important contribution that formalizes a data-inspection approach for corrupted data.

My personal take is that the work is practically useful in many scientific fields where experimental labs lack a framework for deciding whether collecting data from another modality can improve decision-making. I support the acceptance.

**Audience:**

Yes

**Audience Explanation:**

Missing data is ubiquitous in many application domains, particularly in experimental sciences such as healthcare, biology, and sensor-based systems, where data collection is often costly, time-consuming, or constrained by practical limitations. In these settings, large-scale missingness is common, and practitioners frequently face the decision of whether it is worth collecting additional measurements for a given instance. The framework proposed in this paper directly addresses this decision-making problem by quantifying how missing features may affect model predictions.

**Broader Impact Concerns:**

I do not see that the paper raises a broader impact concerns.

**Claims And Evidence:**

Yes

**Claims Explanation:**

While the paper proposes an interesting framework, key aspects of the formulation lack clarity, which weakens the support for its claims.

First, the notation could be made simpler. For instance, the use of the pushforward operator is not intuitive for a general ML audience and is not sufficiently explained. Additionally, the definition of $ \pi(x_m, x_o) $ is ambiguous. Definition 1 suggests a dependence on observed variables $ x_o $, which could be interpreted as $ p(y \mid x_o) $, while Equation (7) implies $ \pi(x_m, x_o) = p(y \mid x_m, x_o) $.

Second, the framework relies on the conditional distribution $ p(x_m \mid x_o) $, but it is not clear how this distribution is obtained in practice. Section 5.1 suggests sampling-based approaches, yet in real-world settings this distribution is typically unknown. The paper does not clearly specify how it is modeled or estimated.

**Requested Changes:**

The paper would benefit from several clarifications and improvements to strengthen its presentation and support its claims.

First, the notation and key concepts should be made more accessible. In particular, the pushforward operator should be explained in an intuitive manner for a general machine learning audience (e.g., as propagating uncertainty in missing inputs through the model to obtain a distribution over predictions). In addition, the definition of $ \pi(x_m, x_o) $ should be clarified and made consistent throughout the paper.

Second, the paper should clearly specify how the conditional distribution $ p(x_m \mid x_o) $ is obtained or approximated in practice.

---

> ### Author Response · Authors · 2026-06-03
> **Author Response**
>
> > **Definition 1** (Missing Value Uncertainty Distribution).
> > *Given a joint distribution $p(X,Y)$, let us define the true class distribution $\pi(x)$ given complete inputs and the corresponding random variable $\Phi$ as:*
> >
> > $\pi(x) \triangleq P(Y|X=x)$, $\Phi \triangleq \pi(X) \equiv \pi(X_o, X_m)$
> >
> > *where $\pi: \mathcal{X} \to \Delta^{|\mathcal{Y}|-1}$ is a deterministic function mapping a complete input to a probability vector on the simplex and $\Phi$ is the random variable representing these class probabilities given the random input $X$, which can be divided into random observed features $X_o$ and missing $X_m$.*
> > *Given these, the Missing Value Uncertainty (MVU) distribution given an observed set of input values $X_0$ (and set of missing input values $X_m$) is defined as the conditional of $\Phi$ given our observation $p(\Phi | X_o = x_o)$.*
>
> Based on your comments, we have updated definition 1 to be more clear, which is quoted above. Our responses below contains the justification for the changes and clarify any remaining details.
>
> > "the use of the pushforward operator is not intuitive for a general ML audience and is not sufficiently explained"
>
> We agree that the push-forward operator may be confusing for a more general audience. That statement was meant to add clarity into the construction of the the distribution, but we can see it just added confusion when the distribution was already well defined.
>
> > "the definition of $\pi(x_m,x_o)$ is ambiguous. Definition 1 suggests a dependence on observed variables $x_o$"
>
> $\pi(x)$ is dependent on the full $x$, which includes $x_o$ and $x_m$. In other words, $\pi(x) \equiv \pi(x_m,x_o)$ as all features in $x$ are either missing or observed.
> $\pi$ is then one of the building blocks for $p(\Phi|x_o)$ as $\Phi = \pi(X)$; we never evaluate $\pi$ directly using just $x_o$, though the input $X_m$ is typically a random variable making $\Phi$ random.
> % We updated definition 1 to directly state $\pi(x) \equiv \pi(x_m,x_o)$ to make this more clear.
>
> > "Second, the framework relies on the conditional distribution $p(x_m|x_o)$, but it is not clear how this distribution is obtained in practice."
>
> It is true that the generator $p(X_m|x_o)$ appears in definition 1 equation 6, and some of our baselines model it, though it is not directly modeled in most of our work. Notably, DMV does not model the generator, instead modeling $p(\Phi|X_o)$ directly.
> The Monte Carlo approximation (section 5.1) samples $p(X_m|x_o)$, which as described is approximated by either a multivariate normal (which we use in our synthetic experiment) or a diffusion model with image inpainting (which we use in our CelebA experiments).
>
> The change to definition 1 makes this more clear as it no longer references the generator. We will additionally clarify this by restating the distributions our models approximate in the experiment description in section 7, notably including DMV (approximates $p(\Phi|X_o))$) and the diffusion (approximates $p(X_m|x_o)$). We will also clarify which distributions are being approximated in section 5.1, as the discussion of efficiency of the Monte Carlo estimate may add some confusion about which distribution the diffusion approximates.

---

> > ### Author Response · Authors · 2026-06-19
> > **Updated Paper Revision**
> >
> > The draft posted earlier this week updates definition 1 to make the relationship between our prediction $\Phi$ and the observed/unobserved features, which we believe resolves your concerns. Does this new draft make our claims and our justifications of them more clear?
> >
> > Since we are nearing the end of the interaction period, we may not be able to directly respond to questions after today.

---

### Review · Reviewer_xBbM · 2026-04-01

**Summary Of Contributions:**

This paper analyzes the Missing Value Uncertainty (MVU) problem, that is, quantifying the uncertainty introduced by missing input features at inference time. The authors formalize this through “hard confidence”: the probability that the prediction remains unchanged after observing missing features, as distinct from “soft confidence”. They propose the Direct Missing Value (DMV) estimator, which learns the MVU distribution from complete training data without conditional sampling. They also introduce a new evaluation metric, the Missing Value Calibration Error (MVCE), and a post-hoc calibration procedure. Experiments are conducted on both synthetic and real datasets to validate their method.

**Additional Comments:**

Section 3 would benefit from engaging more deeply with the Bayesian decision theory literature: soft/hard voting distinction parallels well-known techniques in Bayesian model averaging and decision theory, and claiming that hard voting is "under-explored" might be an overstatement. As far as I understand, for the 0-1 loss, the Bayes-optimal decision under the marginal predictive is soft voting, and the probability that a random draw from the posterior would yield the same decision is hard confidence, which are natural tools for the Bayesian community.

As far as I understood, all the experiments are carried out in the MCAR scenario, which as you acknowledge, is the simplest one. Why not try the MAR or MNAR missingness? Does the DMV estimator depend on the type of missingness? At least a discussion of these points is necessary.

**Audience:**

Yes

**Audience Explanation:**

The paper is well motivated and addresses an important practical challenge. It might be interesting for a broad machine learning and statistical audience.

**Claims And Evidence:**

Yes

**Claims Explanation:**

The DMV method is technically correct, it has practical scalability benefits and is validated with reasonable experiments. The MVCE metric is also a nice contribution.

**Requested Changes:**

Typos: page 8, "relaionship"; pages 11,19, "independant".

---

> ### Author Response · Authors · 2026-06-03
> **Author Response**
>
> > "The paper is well motivated and addresses an important practical challenge. It might be interesting for a broad machine learning and statistical audience."
>
> Your justification here seems to contradict selecting "no" for the paper fitting TMLR's audience. Did you perhaps select the wrong option?
>
> > "Section 3 would benefit from engaging more deeply with the Bayesian decision theory literature: soft/hard voting distinction parallels well-known techniques in Bayesian model averaging and decision theory, and claiming that hard voting is "under-explored" might be an overstatement."
>
> We agree that our claim of hard-confidence being under-explored was an overstatement. We will relax that claim to instead state that it has not been considered in light of the missing value problem, as traditional methods handling missing values are most accurately described as taking the mean of this distribution (providing at most soft confidence).
> In our updated draft, we will better connect section 3 and in particular hard voting to relevant Bayesian literature.
>
> > "As far as I understood, all the experiments are carried out in the MCAR scenario, which as you acknowledge, is the simplest one. Why not try the MAR or MNAR missingness? Does the DMV estimator depend on the type of missingness? At least a discussion of these points is necessary."
>
> We should emphasize that when it comes to missingness, there are three different times it can apply: missingness in training data, simulated missingness during training, and missingness in testing data - see the second paragraph in section 2 under "Missing Values" for more discussion on these.
>
> In our work, we assume no missingness in training data.
> However, DMV training *simulates* missingness using an MCAR setup. We choose MCAR as its the simplest choice without knowledge of the test time missingness mechanism, but this is not a requirement to train the model.
> If we had knowledge of the test time mechanism (even incomplete knowledge) we could incorporate that into the simulated missingness for DMV, and it should learn the MNAR correlations to leverage for predictions.
> However, without any knowledge of the test time mechanism, using MNAR is more likely to add biases that may not exist at test tine.
>
> At test time, we have no knowledge of the missing mechanism used. Additionally, since we assume each sample is classified independently, we have only a single sample which is too little to identify the missingness mechanism.
> Thus, without any knowledge of the test time mechanism, we are limited to using the model we created from training data without fine-tuning.
> Assuming we used MNAR data at test time, we expect DMV to perform well if the simulated missingness is on the same MNAR mechanism. If the simulated mechanism differs too much from the test mechanism, we expect DMV trained on MCAR data to outperform DMV trained on the "wrong" MNAR.
>
> As part of the updated draft, we ran an MNAR experiment to validate the above hypothesis. We trained three DMV models with the Starcraft dataset on different simulated missingness: one on MCAR and two on different MNAR setups ("high" is more likely to be missing when the region has units, while "low" is more likely to be missing when the region has no units). We then evaluated each of the three models on each of the three mutators.
> From this, we verified that DMV does not require MCAR data to train, as the each MNAR model performed better on the matching setup than the MCAR model.
> This also provided evidence for our hypothesis that MCAR produces better results than a MNAR model that was tested on a different MNAR setup; without knowledge of the test mechanism using simulated MCAR in training gives better average case performance than simulated MNAR.
> You can see these results in the tables below (which will be formatted better in the paper). We have updated the paper with this experiment, relocating relevant discussion from section 8 to that experiment.
>
> > Table 1: Evaluation of Test Time Accuracy under models trained by different mutators. Best values bolded, second best italic.
>
> | Training Mutator | Clean | MCAR | MNAR (high) | MNAR (low) |
> |---|---|---|---|---|
> | MCAR        | 79.78% | **79.78%** |  *75.09%*  |  *78.86%*  |
> | MNAR (high) | 80.18% |   76.51%   | **76.18%** |   76.49%   |
> | MNAR (low)  | 80.63% |  *79.25%*  |   71.43%   | **80.17%** |
>
> > Table 2: Evaluation of Test Time MVCE under models trained by different mutators. Best values bolded, second best italic.
>
> | Training Mutator | MCAR | MNAR (high) | MNAR (low) |
> |---|---|---|---|
> | MCAR        | **0.0201** |  *0.0863*  | **0.0090** |
> | MNAR (high) |   0.0412   | **0.0529** |   0.0259   |
> | MNAR (low)  |  *0.0407*  |   0.1358   |  *0.0102*  |

---

> > ### Comment · Reviewer_xBbM · 2026-06-05
> >
> > Sorry for the mistake, obviously I wanted to tick "Yes".
> > Thank you for the answers. I'll be happy to suggest acceptance provided this discussion is incorporated to the new draft.

---

> > > ### Author Response · Authors · 2026-06-19
> > > **Updated Paper Available**
> > >
> > > We believe we have addressed your concerns, incorporating our discussion in the new draft posted earlier this week. Let us know if there are any remaining concerns.
> > >
> > > As we are nearing the end of the interaction period, we may not be able to directly reply to any concerns raised after today.

---

### Review · Reviewer_GbgD · 2026-06-05

**Summary Of Contributions:**

The paper addresses uncertainty arising from missing input features at inference time and asks whether obtaining the missing values would change a model’s prediction. It formalizes this as MVU, distinguishes between standard soft confidence and prediction-stability-based hard confidence, and proposes DMV, an efficient method for estimating this uncertainty without costly sampling or imputation. The paper also introduces MVCE to evaluate the calibration of hard confidence.

Strengths
- Identifies a practical and underexplored decision-making problem related to uncertainty from missing input values.
- Introduces an intuitive and useful distinction between soft confidence and hard confidence.

Weaknesses
- The setting is somewhat restricted, as the method assumes fully observed training data.
- Experiments primarily rely on simulated missingness patterns rather than naturally occurring missing data. The empirical validation is limited in scope, focusing on relatively simple masking schemes and image classification benchmarks.
- It remains unclear how well the approach generalizes to mission-critical sensor, medical, or tabular domains where missingness mechanisms may be structured, costly, or non-random.
- The practical question of whether it is worth collecting missing values is only approximated through prediction stability.
- The paper does not fully evaluate cost-sensitive decision-making or compare against broader active feature acquisition strategies.

**Audience:**

Yes

**Audience Explanation:**

At least some individuals in TMLR’s audience would likely be interested in the findings of this paper. The problem of making reliable predictions under missing input features is broadly relevant to machine learning, especially in settings involving uncertainty estimation, robustness, calibration, active feature acquisition, and decision-making under partial information. The paper’s distinction between soft confidence and hard confidence is also likely to be of interest, since it reframes uncertainty not only as “is the prediction correct?” but also as “would the prediction change if more information were collected?”

**Claims And Evidence:**

Yes

**Claims Explanation:**

The submission’s main claims are supported by generally accurate, clear, and convincing evidence, particularly for the proposed formulation of MVU, the DMV method, the MVCE metric, and the efficiency advantages of DMV relative to sampling-based approaches. The theoretical framework is well motivated, and the experiments demonstrate that DMV can effectively estimate prediction-stability uncertainty while achieving substantially lower computational cost than diffusion-based sampling methods.

However, the evidence is less convincing for broader practical claims. The evaluation relies primarily on simulated missingness patterns and assumes fully observed training data, making it unclear how well the approach generalizes to realistic settings with naturally occurring or structured missingness. Additionally, DMV is not uniformly superior to all baselines across all metrics and datasets. Therefore, while the core technical claims are supported, stronger evidence would be needed to fully substantiate the broader practical implications.

**Requested Changes:**

- The authors should add experiments with more realistic or structured missingness, or at least provide a more systematic sensitivity analysis showing how DMV behaves under different missingness mechanisms and under mismatch between training-time masking and test-time missingness.
- Provide a more explicit discussion of the fully observed training-data assumption. This assumption is central to DMV and MVCE, but it substantially limits applicability to many real missing-data settings
- Some experiments suggest that missing-robust classifiers can perform very well when prediction accuracy remains high under missingness. A more detailed discussion of the tradeoff between improving robustness and estimating missing-value uncertainty would help readers understand the practical use cases of DMV.

---

> ### Author Response · Authors · 2026-06-16
> **Author Response, request changes 1 and 3**
>
> Replying to requested changes 1 and 3, hit a character limit in the slightly longer answer to 2.
>
> > "The authors should add experiments with more realistic or structured missingness, or at least provide a more systematic sensitivity analysis showing how DMV behaves under different missingness mechanisms and under mismatch between training-time masking and test-time missingness."
>
> We created a new experiment that makes use of MNAR structured missingness, which is used to compare DMV under different runtime mechanisms including mismatch between training time and test time missingness. This experiment is included in the updated draft of the paper.
> We trained three DMV models on different simulated missingness: one on MCAR and two on different MNAR setups. We then evaluated each of the three models on each of the three mutators.
> From this, we verified that DMV can work with structured data, as the each MNAR model performed better on the matching setup than the MCAR model.
> This experiment also suggests that without knowledge of the test mechanism using simulated MCAR in training gives better average case performance than simulated MNAR.
> You can see these results in the tables below.
>
> > Table 1: Evaluation of Test Time Accuracy under models trained by different mutators. Best values bolded, second best italic.
>
> | Training Mutator | Clean | MCAR | MNAR (high) | MNAR (low) |
> |---|---|---|---|---|
> | MCAR        | 79.78% | **79.78%** |  *75.09%*  |  *78.86%*  |
> | MNAR (high) | 80.18% |   76.51%   | **76.18%** |   76.49%   |
> | MNAR (low)  | 80.63% |  *79.25%*  |   71.43%   | **80.17%** |
>
> > Table 2: Evaluation of Test Time MVCE under models trained by different mutators. Best values bolded, second best italic.
>
> | Training Mutator | MCAR | MNAR (high) | MNAR (low) |
> |---|---|---|---|
> | MCAR        | **0.0201** |  *0.0863*  | **0.0090** |
> | MNAR (high) |   0.0412   | **0.0529** |   0.0259   |
> | MNAR (low)  |  *0.0407*  |   0.1358   |  *0.0102*  |
>
> > "Some experiments suggest that missing-robust classifiers can perform very well when prediction accuracy remains high under missingness. A more detailed discussion of the tradeoff between improving robustness and estimating missing-value uncertainty would help readers understand the practical use cases of DMV."
>
> On the MNIST data, we see the missing robust classifier had such a small drop in accuracy that it did not need to directly estimate MVU to minimize MVCE. We should emphasize that theoretically, it should be possible to train DMV to have the same accuracy under missing values, though since the objective is also modeling uncertainty it would take longer to train.
>
> The two objectives, accuracy and MVU estimation are effectively orthogonal concepts here. Accuracy corresponds to the mean of the prediction while uncertainty corresponds to the variance. You could model this by splitting the DMV outputs into $\phi$ and a confidence scale $c$ such that the Dirichlet parameters $\alpha = c \cdot \phi$; the part of the model producing $\phi$ can then be effectively the same structure as the robust classifier. Accuracy under missing values depends on correctly modeling $\phi$ while MVCE is largely based on modeling $c$.
>
> Thus, the only tradeoff should be if you have a limited training budget, as its easier to train a robust model than a model that also estimates uncertainty. If you know ahead of time that you can make accurate predictions with the expected number of features, you may decide its not worth modeling MVU as you would never determine its worthwhile to collect features. However, even the theoretical best model will eventually become unusable under enough missingness, so if you expect that level of missingness then its worthwhile to be able to quantify when more information might improve the prediction.
>
> We added two new paragraphs to the discussion section that discuss the value of MVU and the cost of MVU in more depth, which touches on this point.

---

> ### Author Response · Authors · 2026-06-16
> **Author Response, requested change 2**
>
> Discussion on requested change 2 ended up a little long, so had to split our response into two comments.
>
> > "Provide a more explicit discussion of the fully observed training-data assumption. This assumption is central to DMV and MVCE, but it substantially limits applicability to many real missing-data settings"
>
> For our work, we assumed fully observed training-data. This is consistent with the assumption often made in the area of active feature acquisition [Aronsson et al., 2025]. While collecting features is assumed to have a cost, without considering the cost of training there is no reason to not collect all features at training time. This is different from the standard missing value setup which lacks a way to collect missing features.
> However, we understand it is not always feasible to collect fully observed data for training. We may for instance have a set of features that can never be collected if they end up missing, or have a limited data collection budget requiring training on partially observed data.
> This is a non-trivial problem to solve, and we feel solving it is not within the scope of this paper. However, we have some preliminary ideas to discuss the problem that could lead to future work to solve it.
>
> If the missing features in the dataset correspond to data that is collectable in our problem setup but it is not feasible to collect them for training, one approach to use DMV and MVCE involves imputing missing values, ideally with a method that considers the label in the imputation to reduce the chance DMV simply mimics the imputation (i.e. $E[X_m|X_o, Y]$). Both DMV and MVCE would then compare the imputed sample to the original with missing values.
> This may be difficult to evaluate if we use the same imputation approach for training and testing data as it biases our estimator to matching our choice of imputation, so ideally we collect missing features on test samples for MVCE even if we cannot collect missing features for training samples.
>
> If we can assume the missing features in the dataset correspond to uncollectable features or to our maximum budget, then we could define $X_d \subseteq X$ as the set of features in the dataset sample, while $X_o$ further drops features such that $X_o \subseteq X_d$. The classification problem is then redefined as predicting $\Phi_d \triangleq p(Y|X_d)$, leading to a "differential MVU" $p(\Phi_d|X_o)$ instead of "absolute MVU" $p(\Phi|X_o)$.
> To train DMV, we would replace the "complete data classifier" $\hat{\pi}_\theta$ with a classifier trained on the incomplete training data $X_d$, then apply additional masking to produce the mutated features $X_o$ for DMV and MVCE.
> For evaluation, the model would receive $X_o$, but MVCE would compare the prediction to that made using $X_d$.
>
> In either setup, we will need additional theory and experiments to make our model and metric work on partial training data, so we leave this to future work.
> We updated the paper draft in section 8 with additional discussion on this limitations, though we currently do not include the specific potential approaches. We can add them to the appendix if you think it is worthwhile, but they are not developed enough to include in the main body as we don't want to confuse readers into believing our model has been tested to work in those settings.
>
> ## References
>
> Linus Aronsson, Arman Rahbar, and Morteza Haghir Chehreghani. A survey on active feature acquisition strategies. arXiv preprint arXiv:2502.11067, 2025.

---

### Author Response · Authors · 2026-06-03
**Regarding Revised Paper**

We have responded directly to existing reviewer comments, but were told to expect another reviewer so are holding off on uploading a revised version of our paper until then.

---

### Author Response · Authors · 2026-06-16
**Updated Paper Revision**

We have uploaded a new revision of the paper. Changes compared to the previous revision have a blue font.

The new content added based on reviews has put us just over 13 pages. If desired, we can make another revision to trim the content back down to 12 pages.

---

### Decision · Action_Editor_3Wii · 2026-07-11

**Recommendation:** Accept as is

**Audience:**

Yes

**Audience Explanation:**

Missing Value Uncertainty is an important topic in modern Machine Learning

**Claims And Evidence:**

Yes

**Claims Explanation:**

The reviewers and I agree that the paper addresses an important and practically relevant problem, and that its main claims are supported by the theoretical development and empirical results. They requested clearer exposition of the MVU formulation, further discussion of the fully observed training data assumption, and stronger evidence regarding structured or mismatched missingness mechanisms. The authors responded constructively by clarifying the definitions and modeling assumptions, adding MCAR/MNAR experiments, and expanding the discussion of limitations and the relationship between robustness and uncertainty estimation. In light of these revisions and the reviewers’ positive final assessments, I recommend acceptance.